# The PI3K and MAPK/p38 pathways control stress granule assembly in a hierarchical manner

Alexander Martin Heberle[1,*], Patricia Razquin Navas[1,2,*], Miriam Langelaar-Makkinje[1], Katharina Kasack[3,4], Ahmed Sadik[5,6], Erik Faessler[7], Udo Hahn[7], Philip Marx-Stoelting[8], Christiane A Opitz[5,9], Christine Sers[3,4], Ines Heiland[10], Sascha Schäuble[7,11], Kathrin Thedieck[1,2,12]

All cells and organisms exhibit stress-coping mechanisms to ensure survival. Cytoplasmic protein-RNA assemblies termed stress granules are increasingly recognized to promote cellular survival under stress. Thus, they might represent tumor vulnerabilities that are currently poorly explored. The translation-inhibitory eIF2α kinases are established as main drivers of stress granule assembly. Using a systems approach, we identify the translation enhancers PI3K and MAPK/p38 as pro-stress-granule-kinases. They act through the metabolic master regulator mammalian target of rapamycin complex 1 (mTORC1) to promote stress granule assembly. When highly active, PI3K is the main driver of stress granules; however, the impact of p38 becomes apparent as PI3K activity declines. PI3K and p38 thus act in a hierarchical manner to drive mTORC1 activity and stress granule assembly. Of note, this signaling hierarchy is also present in human breast cancer tissue. Importantly, only the recognition of the PI3K-p38 hierarchy under stress enabled the discovery of p38's role in stress granule formation. In summary, we assign a new pro-survival function to the key oncogenic kinases PI3K and p38, as they hierarchically promote stress granule formation.

## Introduction

Stress granules are cytoplasmic RNA-protein assemblies, which in a dynamic, reversible process create a non-membranous compartment (Kedersha & Anderson, 2007) that recruits mRNAs and signaling proteins under stress (Kedersha et al, 2013). Thus, stress granules serve as a stress-driven signaling hub (Kedersha et al, 2013; Heberle et al, 2015), which buffers translation and promotes survival (Arimoto et al, 2008; Tsai & Wei, 2010; Thedieck et al, 2013). In recent years, stress granules have emerged as critical determinants of cancer cell survival. Stress granule components are often up-regulated in tumor cells and promote their survival under endogenous and therapeutic stresses (Anderson et al, 2015; Heberle et al, 2015). Stress granule assembly is initiated by a variety of stress signals that stall translation (Heberle et al, 2015). The best known regulators of stress granule assembly are eukaryotic translation initiation factor 2α (eIF2α) kinases (Anderson et al, 2015), which inhibit eIF2α to reduce global cap-dependent translation (Holcik, 2015). The subsequent release of monosomal mRNA bound to noncanonical preinitiation complexes enables the recruitment of RNA-binding proteins leading to stress granule formation (Anderson et al, 2015; Panas et al, 2016).

Next to eIF2α kinases, the serine–threonine kinase mechanistic/mammalian target of rapamycin complex 1 (mTORC1) also has been suggested to impinge on stress granules, with opposite hypotheses on the mode of regulation. On the one hand, mTORC1 inhibition has been proposed to induce stress granules (Hofmann et al, 2012; Panas et al, 2016). mTORC1 is a master activator of translation (Saxton & Sabatini, 2017), and therefore translation arrest through mTORC1 inhibition by small molecule compounds, nutrient deprivation, or stresses has been suggested to promote stress granule formation. mTORC1 is embedded in a network of oncogenic kinases, often up-regulated in tumors (Saxton & Sabatini, 2017). Although triggering tumor metabolism and growth, oncogenic signaling to

[1]Laboratory of Pediatrics, Section Systems Medicine of Metabolism and Signaling, University of Groningen, University Medical Center Groningen, Groningen, The Netherlands   [2]Department for Neuroscience, School of Medicine and Health Sciences, Carl von Ossietzky University Oldenburg, Oldenburg, Germany   [3]Laboratory of Molecular Tumor Pathology, Institute of Pathology, Charité Universitätsmedizin Berlin, Berlin, Germany   [4]German Cancer Consortium (DKTK), Partner Site Berlin, German Cancer Research Center (DKFZ), Heidelberg, Germany   [5]Brain Cancer Metabolism Group, German Cancer Research Center (DKFZ), German Cancer Consortium (DKTK), Heidelberg, Germany   [6]Faculty of Bioscience, Heidelberg University, Heidelberg, Germany   [7]Jena University Language and Information Engineering Lab, Friedrich-Schiller-University Jena, Jena, Germany   [8]German Federal Institute for Risk Assessment, Strategies for Toxicological Assessment, Experimental Toxicology and ZEBET, German Centre for the Protection of Laboratory Animals (Bf3R), Berlin, Germany   [9]Neurology Clinic and National Center for Tumor Diseases, University Hospital of Heidelberg, Heidelberg, Germany   [10]Faculty of Bioscience, Fisheries and Economics, Department of Arctic and Marine Biology, UiT The Arctic University of Norway, Tromsø, Norway   [11]Systems Biology and Bioinformatics, Leibniz Institute for Natural Product Research and Infection Biology, Hans Knöll Institute, Jena, Germany   [12]Institute of Biochemistry and Center for Molecular Biosciences Innsbruck, University of Innsbruck, Innsbruck, Austria

Correspondence: ines.heiland@uit.no; sascha.schaeuble@uni-jena.de; sascha.schaeuble@hki-jena.de; kathrin.thedieck@uibk.ac.at; k.thedieck@umcg.nl; kathrin.thedieck@uni-oldenburg.de
*Alexander Martin Heberle and Patricia Razquin Navas contributed equally to this work

mTORC1 might counteract stress granule assembly, thereby reducing tumor cell survival.

In contrast, other reports suggest that mTORC1 enhances stress granule assembly through phosphorylation of its substrates ribosomal protein S6 kinase β-1 (p70-S6K) (Sfakianos et al, 2018) and eukaryotic translation initiation factor 4E-binding protein 1 (4E-BP1) (Fournier et al, 2013). This would suggest that rather than mTORC1 inhibition, its activation under stress is required to enable stress granule formation. If so, stress signals, which activate mTORC1, could not only enhance tumor cell growth but also survival. Thus, inhibitors of the kinase network converging on mTORC1, used as targeted drugs in cancer therapy (Porta et al, 2014; Saxton & Sabatini, 2017), might disrupt stress granules. Therefore, stress granules could serve as a novel surrogate marker for therapy response.

If mTORC1 activity is needed for stress granule assembly, this implies that mTORC1 is active under stress. Inhibitory stress inputs to mTORC1 have been reported in much detail (Demetriades et al, 2016; Saxton & Sabatini, 2017; Su & Dai, 2017), but only little is known about activating stress signals to mTORC1 (Heberle et al, 2015; Su & Dai, 2017). Furthermore, it is unknown whether such activating signaling cues inhibit or promote stress granule assembly. To tackle this gap in our knowledge, we used a combined experimental and computational modelling approach to systematically identify kinases, which activate mTORC1 upon stress. We identify the oncogenic kinases phosphoinositide 3-kinase (PI3K) and MAPK p38 (MAPK/p38) as main stress-driven activators of mTORC1. We also addressed their hierarchy and found that when highly active, PI3K is the main driver of mTORC1 activity; however, the impact of p38 becomes apparent as PI3K activity declines.

Importantly, we report that both PI3K and p38 promote stress granule formation. So far, translation-inhibitory cues have been recognized to mainly drive stress granule assembly (Panas et al, 2016; Protter & Parker, 2016). In contrast, we establish here that the two major translation activators, PI3K (Robichaud & Sonenberg, 2017; Saxton & Sabatini, 2017) and p38 (Gonskikh & Polacek, 2017; Robichaud & Sonenberg, 2017), enhance stress granule assembly by activating mTORC1 under stress. Thus, we assign a new function to the key oncogenic kinases PI3K and p38: not only do they promote tumors by enhancing their metabolism and growth but they also concomitantly enhance stress granule formation, likely increasing tumor cell survival. We coin the term of pro-stress-granule-kinases for them and we anticipate that future studies will add further oncogenic kinases to this family.

# Results

## mTORC1 activity is required for stress granule formation

The carcinogenic metalloid arsenite is a frequently used tool compound to study stress granules (Kedersha & Anderson, 2007; Anderson et al, 2015; Turakhiya et al, 2018). Arsenite induces acute stress by interference with thiol groups of numerous proteins, such as, glutathione, glutathione reductase, and thioredoxin reductase, thus leading to oxidative stress and ultimately translation arrest

(Hughes, 2002; McEwen et al, 2005). In agreement with previous studies (White et al, 2007; Fournier et al, 2013), we found that in MCF-7 breast cancer cells, arsenite exposure led to stress-induced translation arrest, as determined by the phosphorylation of eIF2α at serine 51 (eIF2α-S51) (Fig 1A and B). Furthermore, arsenite induced stress granule assembly (Fig 1C), as monitored by the appearance of cytoplasmic foci, positive for the bona fide stress granule marker G3BP1 (Ras GTPase-activating protein-binding protein 1). Arsenite exposure did not affect G3BP1 levels (Fig S1A and B), indicating that the differences in immunofluorescence were due to granule localization.

Although inhibitory or activating effects of arsenite on mTOR have been reported (Chen & Costa, 2018), we found in MCF-7 breast cancer cells that arsenite stress activated mTORC1, as monitored by the phosphorylation of p70-S6K at threonine 389 (p70-S6K-T389) and 4E-BP1 at threonine 37/46 (4E-BP1-pT37/46) (Fig 1A and B). Note that the 4E-BP1 protein is present in three forms with different phosphorylation states. The hypophosphorylated α and β forms run at a lower molecular size than the hyperphosphorylated γ form (Yamaguchi et al, 2008). All three forms can be phosphorylated by mTORC1 at T37/46, and thus phosphorylation of all 4E-BP1 forms has to be considered to assess mTORC1 activity. Stress-induced phosphorylation of p70-S6K-T389 and 4E-BP1-T37/46 was mediated by mTORC1, as both events were prevented by the allosteric mTORC1 inhibitor everolimus (Sedrani et al, 1998; Kirchner et al, 2004), and by the ATP-competitive mTOR inhibitor AZD8055 (Chresta et al, 2010) (Fig 1D and E).

mTORC1 inhibition (Hofmann et al, 2012; Panas et al, 2016) or activity (Fournier et al, 2013; Sfakianos et al, 2018) has been suggested to mediate stress granule assembly. We found that both everolimus and AZD8055 decreased the numbers of G3BP1-positive foci without affecting G3BP1 levels, suggesting that stress granule formation was inhibited (Fig 1D, F, and G). Thus, we conclude that mTORC1 activity is required for stress granule formation.

## PI3K promotes stress granule formation

To investigate which pathways promote stress granule assembly through mTORC1, we first monitored the stress-induced phosphorylation dynamics across the network upstream of mTOR. For this purpose, we performed time course experiments upon arsenite stress for up to 60 min (Fig S1C and D). Already after 5 min, arsenite acutely enhanced phosphorylation of the AGC kinase Akt at T308 (Akt-T308), which remained high throughout the time course. Class I PI3Ks have been proposed to mediate Akt activation upon stress (Chen & Costa, 2018). In agreement, stress-induced Akt-T308 phosphorylation was prevented by the pan-PI3K inhibitor wortmannin (Arcaro & Wymann, 1993) and the class I PI3K-specific inhibitor GDC-0941 (Folkes et al, 2008) (Fig 2A and B). Upon insulin, the phosphoinositide-dependent protein kinase PDK1 mediates Akt activation downstream of PI3K, but it is unknown whether this is also the case under stress. We found that the PDK1 inhibitor GSK2334470 (Najafov et al, 2011) blocked stress-induced Akt-T308 and reduced p70-S6K-T389 phosphorylation (Fig 2C and D), demonstrating that PDK1 transduces stress signals to Akt and mTORC1.

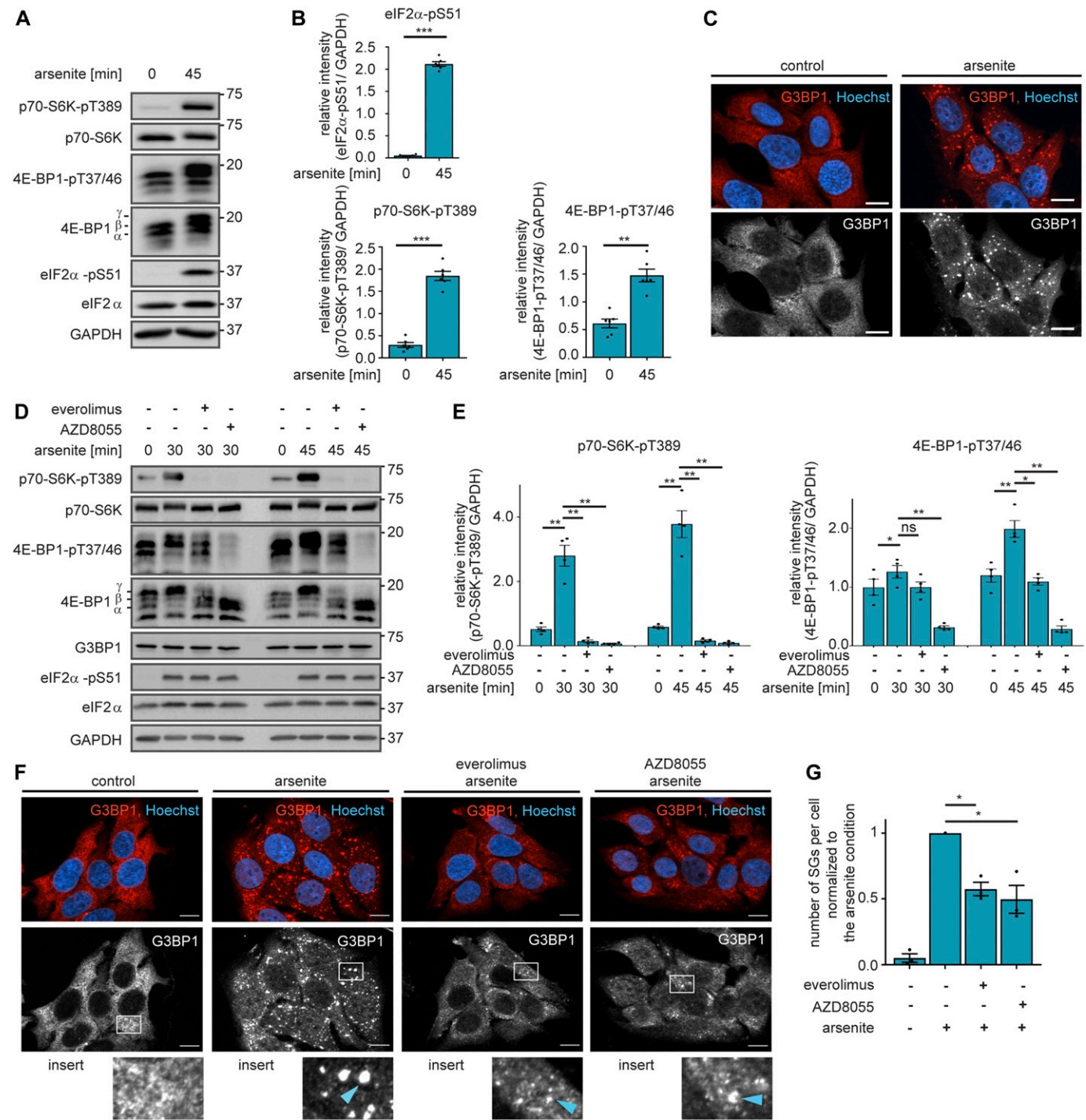

**Figure 1. Stress activates mTORC1 to promote stress granule formation.**
**(A)** Arsenite stress enhances phosphorylation of eIF2α-S51 and mTORC1 substrates. MCF-7 cells were serum-starved and treated with arsenite. p70-S6K-pT389, 4EBP1-pT37/46, and eIF2α-pS51 were monitored by immunoblot. Data represent six biological replicates. **(B)** Quantification of data shown in (A). eIF2α-pS51, p70-S6K-pT389, and 4E-BP1-pT37/46 levels were compared between control and arsenite-treated cells using a two-tailed $t$ test across six biological replicates. Data represent the mean ± SEM. **$P \leq 0.01$; ***$P \leq 0.001$. **(C)** Arsenite induces stress granules. MCF-7 cells were serum-starved and exposed to arsenite for 30 min. Stress granules were visualized by immunofluorescence staining of G3BP1. Data represent three biological replicates. Nuclei were visualized using Hoechst 33342. Scale bar: 10 μm. **(D)** mTOR mediates induction of p70-S6K-pT389 and 4EBP1-pT37/46 by stress. MCF-7 cells were serum-starved and treated with arsenite in the presence of carrier (DMSO), everolimus (100 nM, mTORC1 inhibitor), or AZD8055 (100 nM, mTOR inhibitor). p70-S6K-pT389, 4E-BP1-pT37/46, G3BP1, and eIF2α-pS51 were monitored by immunoblot. Data represent four biological replicates. **(E)** Quantification of data shown in (D). p70-S6K-pT389 and 4E-BP1-pT37/46 levels were compared between the different treatments using a two-tailed $t$ test across four biological replicates. Data represent the mean ± SEM. ns, not significant; *$P \leq 0.05$; **$P \leq 0.01$. **(F)** mTOR inhibition reduces stress granule numbers. MCF-7 cells were serum-starved and treated with arsenite for 30 min in the presence of carrier (DMSO), everolimus (100 nM, mTORC1 inhibitor), or AZD8055 (100 nM, mTOR inhibitor). Stress granules were

In the presence of growth factors and nutrients, Akt inhibits the heteromeric tuberous sclerosis complex (TSC) (Switon et al, 2016) by phosphorylating its component TSC2 at T1462 (Manning et al, 2002). Consequently, the TSC complex detaches from lysosomes (Cai et al, 2006; Menon et al, 2014), the platform at which mTORC1 is active (Saxton & Sabatini, 2017). The TSC is the GTPase-activating protein for the small GTPase Rheb, which activates mTORC1 at the lysosome (Manning & Cantley, 2003). Thus, the TSC acts as an insulin and amino acid–sensitive lysosomal mTORC1 suppressor (Demetriades et al, 2014; Menon et al, 2014). Under stress, TSC activation and localization to lysosomes has been proposed as a universal response to cellular stress (Demetriades et al, 2016). In contrast, we observed enhanced phosphorylation of TSC2-T1462 upon arsenite stress (Fig S1C and D). Thus, we report that stress-induced Akt inhibits the TSC. PI3K inhibition reduced TSC2-T1462 phosphorylation and p70-S6K-pT389 (Fig 2A and B), demonstrating that under stress, PI3K and Akt signal through the TSC to activate mTORC1. Thus, we conclude that TSC inhibition upon stress contributes to mTORC1 activation. We propose that the TSC acts as an integrator of activating and inhibitory stress cues, rather than being a universal mediator of inhibitory stress signals.

We next asked which upstream cue activates PI3K upon stress. The insulin receptor (IR) has been proposed to enhance PI3K signaling upon oxidative stress (Mehdi et al, 2005). To test this, we inhibited the insulin receptor substrate 1 (IRS1) (Fig 2E and F), required for IR-mediated activation of PI3K (Razquin Navas & Thedieck, 2017). Whereas IRS1 protein levels were significantly reduced by two siRNAs, Akt-T308 and p70-S6K-T389 phosphorylations were not altered (Fig 2E and F). Thus, the IR-IRS1 axis does not enhance PI3K-PDK1 signaling nor mTORC1 activity upon arsenite stress. The super-oncogene RAS is another upstream regulator of PI3K (Rodriguez-Viciana et al, 1994), known to respond to growth factor receptors. RAS is a small GTPase, which is active when GTP-bound. To test whether RAS is activated by arsenite stress, we analysed GTP loading of RAS. For this purpose, we performed pull-down experiments with a GST-coupled RAF-RAS–binding domain (RBD), which specifically binds active, GTP-bound RAS but not inactive, GDP-bound RAS. RBD-bound RAS was detected using a pan-RAS antibody. Arsenite stress enhanced RAS-GTP levels, as determined by increased RBD-bound RAS (Fig 2G and H), whereas total RAS remained constant (Fig S1E and F). Thus, arsenite stress enhances RAS activity, which may promote PI3K signaling to mTORC1.

In summary, we report that stress activation of mTORC1 by PI3K is mediated by PDK1, Akt, and the TSC. Yet, it is unknown whether PI3K affects stress granule assembly. Therefore, we analysed stress granule formation upon arsenite, in the absence or presence of the PI3K inhibitor wortmannin. PI3K inhibition significantly reduced the number of arsenite-induced stress granules (Fig 2I and J). We, therefore, conclude that PI3K is required for stress granule

assembly. Thus, we assign a novel function to PI3K as a driver of stress granule formation, and its pro-stress granule activity might contribute to PI3K's oncogenic capacity.

## Computational modelling predicts PI3K-independent stress signals to Akt and mTORC1

Although PI3K and PDK1 inhibitors led to a decline in p70-S6K-pT389, we observed that also in the presence of these inhibitors p70-S6K-T389 phosphorylation was still significantly inducible by arsenite stress (Fig 2B and D). Also stress granule formation was significantly reduced, but still occurred under PI3K inhibition (Fig 2I and J). This suggests that next to PI3K, there is another stress-induced signaling route, which promotes stress granule assembly.

We went on to identify this cue. In principle, any molecule downstream or independent of PI3K could be the transducer of the additional stress signal(s) (Fig S2A). Rather than testing all possible mediators one by one experimentally, we opted for a systematic computational approach to guide subsequent experimental investigation. In the first attempt to comprehensively model the dynamics of the mTOR network upon stress, we built an ordinary differential equation–based computational model. We designed our modelling strategy to systematically map stress-responsive components across the mTOR network.

The initial network structure was based on literature knowledge on molecules and mechanisms, known to signal to mTOR in response to growth factors and nutrients. We then calibrated the model on dynamic stress-time course data to systematically and comprehensively identify the stress inputs to the network.

The initial network structure without stress inputs was as follows (Fig S2A and B, see the Materials and Methods section for details): amino acids directly activate mTORC1 (Saxton & Sabatini, 2017), which is defined in our model by the phosphorylation of the mTORC1 substrates p70-S6K-T389, 4E-BP1-T37/46, and PRAS40-S183 (proline-rich Akt substrate of 40 kD). Downstream of insulin and the IR, PI3K activates PDK1, which in turn phosphorylates Akt (Akt → Akt-pT308) (Alessi et al, 1997; Stephens et al, 1998). Akt activates mTORC1 by inhibiting its endogenous suppressors TSC and PRAS40 through phosphorylation at TSC2-T1462 (Manning et al, 2002) and PRAS40-T246 (Kovacina et al, 2003; Nascimento et al, 2010), respectively (Akt → TSC2-pT1462; Akt → PRAS40-pT246). Akt and p70-S6K belong to the family of AGC kinases, which are activated by phosphorylation in their activation loop and the hydrophobic motif (Pearce et al, 2010; Leroux et al, 2018). Although the kinase for the activation loop is PDK1, the kinases phosphorylating the hydrophobic motif, historically termed PDK2, differ between AGC kinases (Pearce et al, 2010; Leroux et al, 2018). The second mTOR complex, mTORC2, is considered the main PDK2 for Akt. mTORC2 is structurally and functionally distinct from mTORC1 and, in response to activation by PI3K, phosphorylates Akt at S473 (Manning & Toker, 2017). mTORC1 is the PDK2 for p70-S6K (T389), which is also

---

visualized by immunofluorescence staining of G3BP1. Nuclei were visualized with Hoechst 33342. Data represent three biological replicates. White squares indicate region of insert and blue arrows highlight stress granules; scale bar 10 $\mu$m. **(G)** Quantification of data shown in (F): number of stress granules (SGs) per cell (normalized to the arsenite condition) across three biological replicates. Stress granule numbers were compared between the carrier and everolimus, or carrier and AZD8055-treated cells, using a two-tailed $t$ test across three biological replicates. Data represent the mean ± SEM. *$P \leq 0.05$.

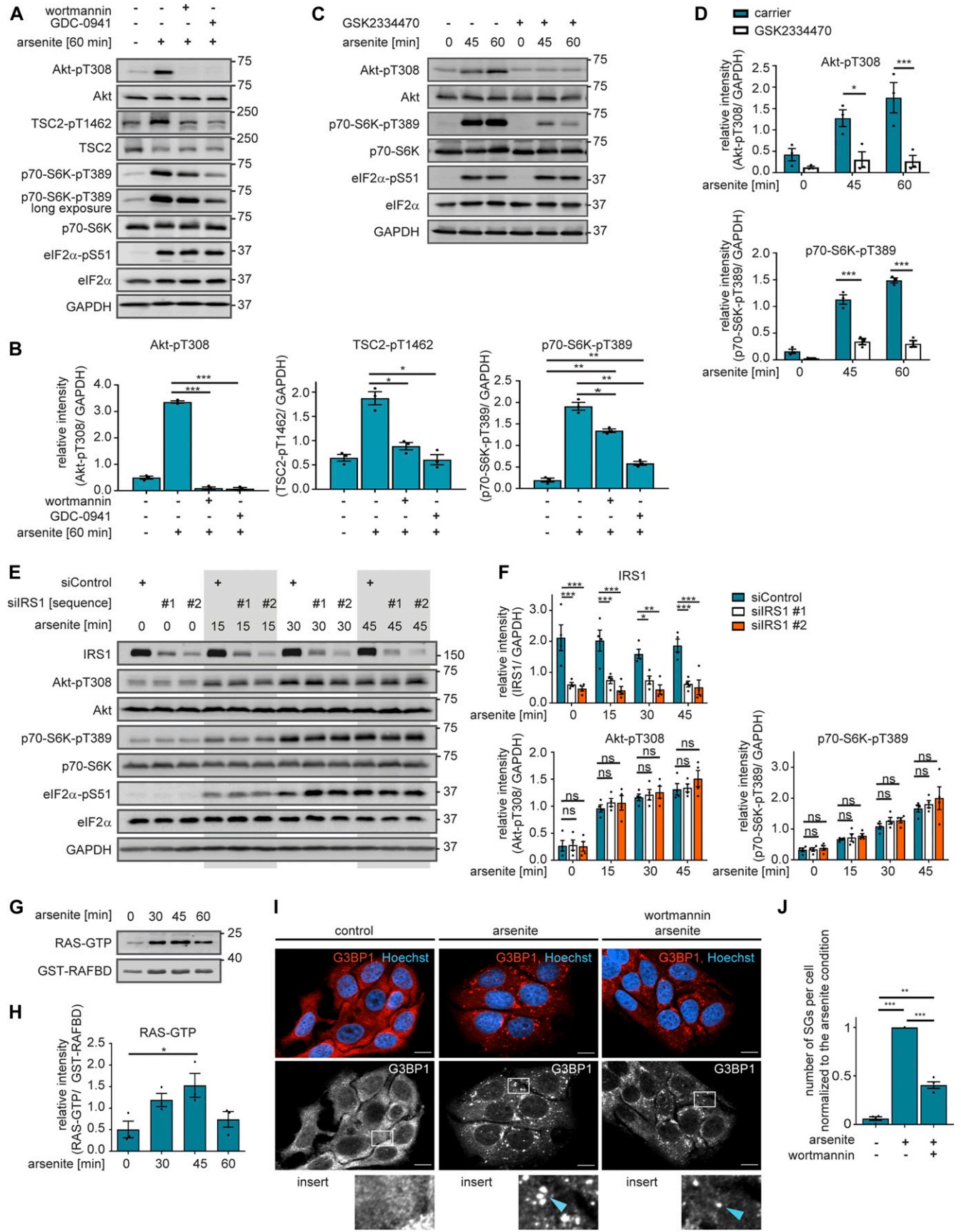

phosphorylated by PDK1 at T229 (Pullen et al, 1998). The two AGC kinase modules in our model include all possible combinations of phosphorylation states, that is, non-phosphorylated, single-, or double-phosphorylated species. All single- and double-phosphorylated Akt species are defined as active in our model, whereby phosphorylation at the PDK1 and 2 sites have additive effects on Akt activity. For p70-S6K in our model, only the species phosphorylated at T389 is defined as active, as p70-S6K activity toward its substrate IRS1-S636/639 has been reported to be mTORC1 dependent (Carlson et al, 2004; Um et al, 2004; Tzatsos & Kandror, 2006).

We calibrated our dynamic model with quantitative time course data recorded for up to 60 min upon arsenite exposure (Fig S1C and D). As we had found that the PI3K-Akt signaling axis transduces a stress signal, which enhances mTORC1 activity and stress granule formation (Fig 2A–D, I, and J), we also recorded perturbation time course datasets, in which we inhibited PI3K (wortmannin) or Akt (MK-2206 [Hirai et al, 2010]) (Fig S3). To include the inhibitors into our model, we calculated the extent of inhibition for wortmannin or MK-2206 by quantifying the relative decrease of Akt-T308 or TSC2-T1462 phosphorylation, respectively, as these are readouts for PI3K and Akt activity (Fig S2C–E). The extent of inhibition was determined as being 100% for wortmannin (modelled as full inhibition) and 83% for MK-2206 (modelled as partial inhibition).

When calibrated on stress time course data, the model with nutritional inputs only (model I, Supplemental Data 1) was not able to fit the measured data for most readouts, including p70-S6K-T389, Akt-T308, TSC2-T1462, p70-S6K-T229, and IRS1-S636/639 phosphorylation (experimental data shown as dots and simulations shown as lines in Fig S4). This formally showed that a model without stress-specific inputs is not sufficient to reproduce the stress response of the network.

Thus, one or several additional inputs were needed for the model to be able to fit the stress time course data. For this purpose, we added, in agreement with our previous findings (Fig 2), a stress input at the level of PI3K (model II, shown in Figs 3A and S5 and Supplemental Data 2), and calibrated this model on our time course data (Figs S1C, D, and S3). Model II was able to simulate the dynamic response of all components across the mTOR network to arsenite stress (Fig 3B, blue; Fig S6A and B). Hence, the addition of a stress input to PI3K improved the fit between the simulations and the data. However, for PI3K perturbation (wortmannin), the model failed to simulate the remaining activity at the level of p70-S6K-pT389 and Akt-pS473 as well as other readouts across the network (Fig 3B, orange; Fig S6C). This confirmed that stress activation of PI3K is not sufficient to explain the dynamic stress response of the mTOR network. Thus, we concluded that additional stress inputs might further improve the fit between the simulated and the measured perturbation data.

To identify further stress-responsive components in the mTOR network, we generated a series of models, in which we systematically included a second stress input to every species across our model II. We compared the fit between the model simulations and the measured data by calculating the Akaike information criterion (AIC) (Table 1). We considered that a model represented an improvement over the previous version if the AIC was reduced by at least 5%. Only two models reached the threshold (Table 1, models 1 and 2), in both of which the second stress input targeted the phosphorylation of Akt at serine 473 (Akt → Akt-pS473 and Akt-pT308 → Akt-pT308-pS473). From these two models, we chose to proceed with model 1 (now renamed model III) as it had the lowest AIC value (1,127) (Figs 3C and S7 and Supplemental Data 3). Model III, with two stress inputs on PI3K and Akt-pS473, correctly simulated the remaining activity at the level of Akt-pS473 and p70-S6K-pT389 under PI3K inhibition (Fig 3D, orange; Fig S8).

In our models I–III, the full Akt module including phosphorylation at T308 and S473 was upstream of mTORC1. Hence, we hypothesized that the improved fit between the p70-S6K-pT389 simulation and measured data in model III might come from the connection between Akt-pS473 and mTORC1. To test whether this was the case, we removed the link between Akt-pS473 and mTORC1 from our model (model IV, Figs S9 and S10 and Supplemental Data 4). Indeed, this model IV failed to simulate the remaining activity of

**Figure 2. PI3K enhances stress granule formation through mTORC1 activation.**
**(A)** mTORC1 is activated by arsenite in a PI3K-dependent manner. MCF-7 cells were serum-starved and treated with arsenite in the presence of carrier (DMSO), wortmannin (100 nM, PI3K inhibitor), or GDC-0941 (1 μM, PI3K inhibitor). Akt-pT308, TSC2-pT1462, p70-S6K-pT389, and eIF2α-pS51 were monitored by immunoblot. Data represent three biological replicates. **(B)** Quantification of data shown in (A). Akt-pT308, TSC2-pT1462, and p70-S6K-pT389 levels were compared between carrier and wortmannin as well as carrier and GDC-0941-treated cells using a two-tailed $t$ test across three biological replicates. Data represent the mean ± SEM. *$P$ ≤ 0.05; **$P$ ≤ 0.01; ***$P$ ≤ 0.001. **(C)** PDK1 mediates stress activation of mTORC1. MCF-7 cells were serum-starved and treated with arsenite in the presence of carrier (DMSO) or GSK2334470 (1 μM, PDK1 inhibitor). Akt-pT308, p70-S6K-pT389, and eIF2α-pS51 were monitored by immunoblot. Data represent three biological replicates. **(D)** Quantification of data shown in (C). Akt-pT308 and p70-S6K-pT389 levels were compared between carrier and GSK2334470-treated cells using a two-way ANOVA followed by a Bonferroni multiple comparison test across three biological replicates. Data represent the mean ± SEM. $P$-values for the Bonferroni multiple comparison tests are shown above the columns. *$P$ ≤ 0.05; ***$P$ ≤ 0.001. **(E)** Stress activation of mTORC1 is IRS1 independent. MCF-7 cells treated with non-targeting scramble siRNA (siControl) or with two different siRNA sequences targeting IRS1 (siIRS1 #1 and #2) were serum-starved and treated with arsenite. Akt-pT308, p70-S6K-pT389, and eIF2α-pS51 were monitored by immunoblot. Data represent four biological replicates. **(F)** Quantification of data shown in (E). IRS1, Akt-pT308, and p70-S6K-pT389 levels were compared between siControl, siIRS1 #1–, and siIRS2 #2–treated cells using a two-way ANOVA followed by a Bonferroni multiple comparison test across four biological replicates. Data represent the mean ± SEM. $P$-values for the Bonferroni multiple comparison tests are depicted above the corresponding time point. ns, not significant; *$P$ ≤ 0.05; **$P$ ≤ 0.01; ***$P$ ≤ 0.001. **(G)** Stress activates RAS. MCF-7 cells were serum-starved and treated with arsenite. RAS activity was measured using GST-coupled RAF-RAS–binding domain pull down experiments. Data represent three biological replicates. **(H)** Quantification of data shown in (G). RAS-GTP levels were compared over an arsenite time course using a one-way ANOVA followed by a Bonferroni multiple comparison test across three biological replicates. Data represent the mean ± SEM. The significances for the Bonferroni multiple comparison tests between time points is shown above the column, the $P$-value for the ANOVA is $P$ = 0.0318. *$P$ ≤ 0.05. **(I)** PI3K inhibition reduces stress granule numbers. MCF-7 cells were serum-starved and treated with arsenite for 30 min in the presence of carrier (DMSO) or wortmannin (100 nM, PI3K inhibitor). Stress granules were visualized by immunofluorescence staining of G3BP1. Nuclei were visualized with Hoechst 33342. Data represent four biological replicates. White square indicates region of insert and blue arrow highlights stress granules; scale bar 10 μm. **(J)** Quantification of data shown in (I): number of stress granules (SGs) per cell (normalized to the arsenite condition) across four biological replicates. A two-tailed $t$ test across four biological replicates was applied. Data represent the mean ± SEM. **$P$ ≤ 0.01; ***$P$ ≤ 0.001.

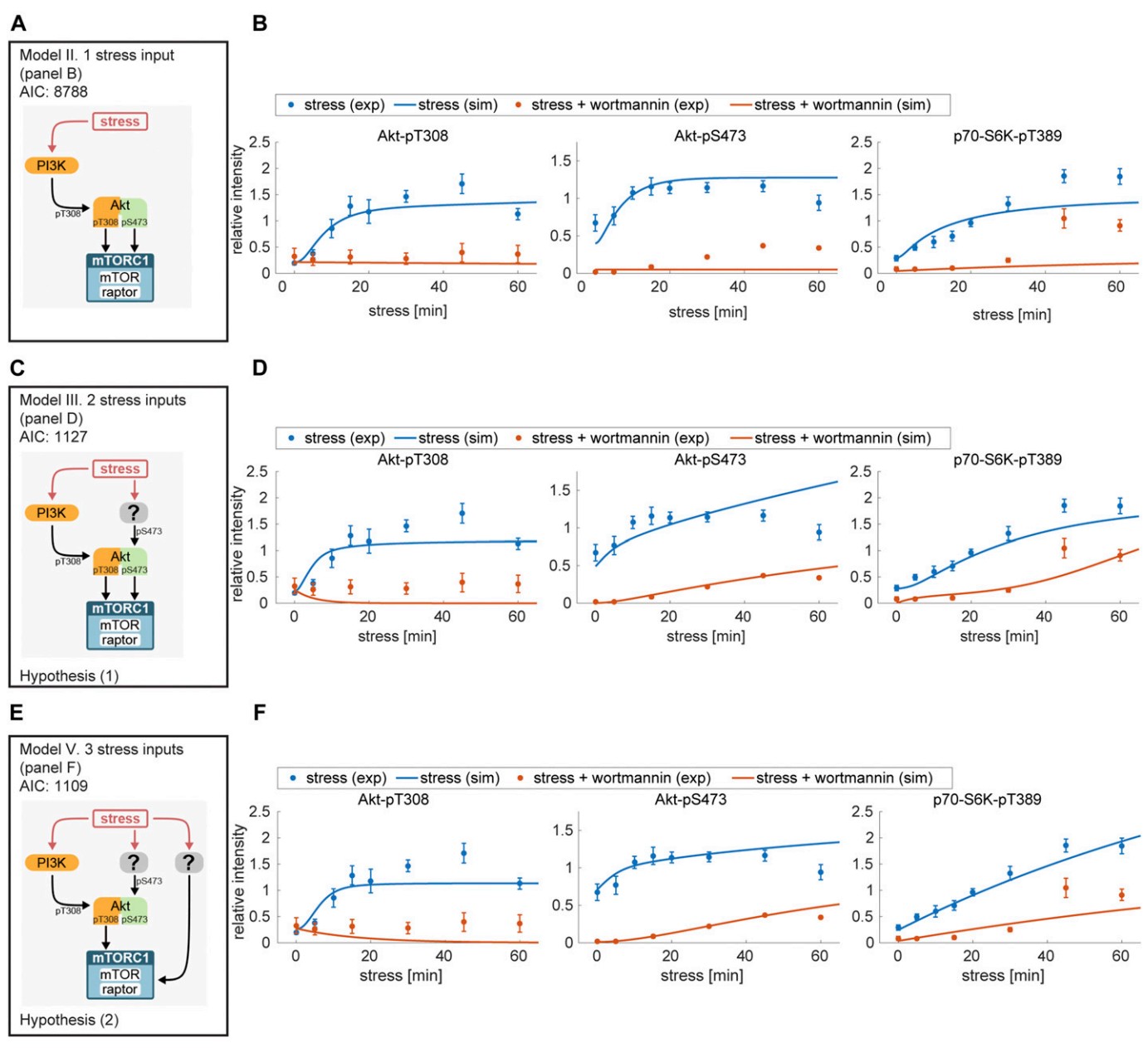

**Figure 3. Computational model predicts three stress inputs on PI3K, Akt, and mTORC1.**
**(A)** Scheme of model II, with only one stress input on PI3K (Fig S5). The corresponding AIC value is indicated at the top. **(B)** Simulated response of Akt-pT308, Akt-pS473, and p70-S6K-pT389 to arsenite stress in a system without (blue) or with (orange) wortmannin (PI3K perturbation) using model II, with one stress input on PI3K. Blue dots represent experimental data from arsenite time course (Fig S1C and D) and orange dots represent experimental data from arsenite time course + PI3K perturbation (Fig S3A and B). Lines represent computational simulation. Data represent the mean ± SEM. Simulations of all used observables are shown in Fig S6. **(C)** Scheme of model III, with two stress inputs on PI3K and Akt-pS473 (Fig S7). The corresponding AIC value is indicated at the top. **(D)** Simulated response of Akt-pT308, Akt-pS473, and p70-S6K-pT389 to arsenite stress in a system without (blue) or with (orange) wortmannin (PI3K perturbation) using model III, with two stress inputs on PI3K and Akt-pS473. Dots represent experimental data and lines represent computational simulation as described in (B). Data represent the mean ± SEM. Simulations of all used observables are shown in Fig S8. **(E)** Scheme of model V, with three stress inputs on PI3K, Akt-pS473, and mTORC1 (Fig S11). The corresponding AIC value is indicated at the top. **(F)** Simulated response of Akt-pT308, Akt-pS473, and p70-S6K-pT389 to arsenite stress in a system without (blue) or with (orange) wortmannin (PI3K perturbation) using model V, with three stress inputs on PI3K, Akt-pS473, and mTORC1. Dots represent experimental data and lines represent computational simulation as described in (B). Data represent the mean ± SEM. Simulations of all used observables are shown in Fig S12.

p70-S6K-pT389 under PI3K perturbation (Fig S10C). Thus, the fit between the p70-S6K-pT389 simulations and measured data in model III is mediated by Akt-pS473. In agreement, it has been proposed that Akt-S473 phosphorylation can enhance mTORC1

activity (Hart & Vogt, 2011). Yet, in contrast, several studies have claimed that there is no positive link between mTORC1 and mTORC2 (Guertin et al, 2006; Jacinto et al, 2006), the main mediator of Akt-S473 phosphorylation.

**Table 1. AIC values for input search.**

| Model no. | Substrate | Product | AIC | AICc | BIC |
|---|---|---|---|---|---|
| Single stress input | | | | | |
| II | PI3K | PI3K_p | 8,788 | 8,844 | 9,126 |
| Second stress input in addition to PI3K | | | | | |
| 1 | Akt | Akt_pS473 | 1,127 | 1,185 | 1,469 |
| 2 | Akt_pT308 | Akt_pT308_pS473 | 1,535 | 1,593 | 1,877 |
| 3 | PDK1_cyt | PDK1_mem | 8,595 | 8,653 | 8,937 |
| 4 | Akt | Akt_pT308 | 8,597 | 8,655 | 8,939 |
| 5 | TSC2 | TSC2_pT1462 | 8,599 | 8,657 | 8,941 |
| 6 | PRAS40 | PRAS40_pT246 | 8,623 | 8,681 | 8,965 |
| 7 | PRAS40 | PRAS40_pS183 | 8,663 | 8,721 | 9,005 |
| 8 | p70_S6K | p70_S6K_pT389 | 8,667 | 8,724 | 9,009 |
| 9 | PRAS40_pS183 | PRAS40_pT246_pS183 | 8,704 | 8,761 | 9,045 |
| 10 | p70_S6K | p70_S6K_pT229 | 8,706 | 8,763 | 9,048 |
| 11 | IR_beta | IR_beta_pY1146 | 8,733 | 8,791 | 9,075 |
| 12 | IRS1 | IRS1_loc_pS636 | 8,734 | 8,791 | 9,076 |
| 13 | IRS1 | IRS1_loc | 8,738 | 8,795 | 9,080 |
| 14 | 4EBP1 | 4EBP1_pT37_46 | 8,762 | 8,820 | 9,104 |
| 15 | PRAS40_pS183 | PRAS40 | 8,763 | 8,821 | 9,105 |
| 16 | p70_S6K_pT229 | p70_S6K_pT229_pT389 | 8,768 | 8,826 | 9,110 |
| 17 | p70_S6K_pT389 | p70_S6K_pT229_pT389 | 8,776 | 8,834 | 9,118 |
| 18 | Akt_pT308_pS473 | Akt_pT308 | 8,777 | 8,835 | 9,119 |
| 19 | Akt_pT308_pS473 | Akt_pS473 | 8,781 | 8,839 | 9,123 |
| 20 | PRAS40_pT246 | PRAS40_pT246_pS183 | 8,781 | 8,839 | 9,123 |
| 21 | IRS1_loc | IRS1_loc_pS636 | 8,781 | 8,839 | 9,123 |
| 22 | TSC2_pT1462 | TSC2 | 8,781 | 8,839 | 9,123 |
| 23 | PI3K_p | PI3K | 8,781 | 8,839 | 9,123 |
| 24 | PDK1_mem | PDK1_cyt | 8,781 | 8,839 | 9,123 |
| 25 | p70_S6K_pT229 | p70_S6K | 8,781 | 8,839 | 9,123 |
| 26 | IR_beta_pY1146 | IR_beta | 8,782 | 8,840 | 9,124 |
| 27 | Akt_pS473 | Akt_pT308_pS473 | 8,783 | 8,841 | 9,125 |
| 28 | PRAS40_pT246 | PRAS40 | 8,788 | 8,846 | 9,130 |
| 29 | PRAS40_pT246_pS183 | PRAS40_pS183 | 8,788 | 8,846 | 9,130 |
| 30 | 4EBP1_pT37_46 | 4EBP1 | 8,790 | 8,848 | 9,132 |
| 31 | PRAS40_pT246_pS183 | PRAS40_pT246 | 8,790 | 8,848 | 9,132 |
| 32 | p70_S6K_pT229_pT389 | p70_S6K_pT389 | 8,791 | 8,849 | 9,133 |
| 33 | Akt_pS473 | Akt | 8,791 | 8,849 | 9,133 |
| 34 | Akt_pT308 | Akt | 8,797 | 8,855 | 9,139 |
| 35 | IRS1_loc_pS636 | IRS1 | 8,799 | 8,857 | 9,141 |
| 36 | p70_S6K_pT389 | p70_S6K | 8,799 | 8,857 | 9,141 |
| 37 | p70_S6K_pT229_pT389 | p70_S6K_pT229 | 8,803 | 8,861 | 9,145 |

Model fitting quality determined by computing the AIC, the AIC corrected for small sample sizes (AICc), or the BIC. Upper part: fitting quality of model II. Lower part: fitting quality of model II with an additional stress input. Each value is computed by running Latin hypercube sampling 500 times with the indicated stress inputs.

Therefore, it is unknown whether under stress, Akt-pS473 is upstream of mTORC1. Thus, we postulated two hypotheses: when PI3K is inhibited, (1) stress-induction of Akt-pS473 mediates mTORC1 activation (Fig 3C) and in this case, model III is correct; or (2) Akt-S473 phosphorylation does not activate mTORC1 and there is an additional stress input to mTORC1 (Fig 3E). If hypothesis (2) is correct, the addition of this third input to our model would be expected to further improve the fit between the model simulations and the measured data. To test the two hypotheses in silico, we added a third stress input to mTORC1 into model IV (model V, Figs 3E and S11 and Supplemental Data 5). Model V with stress inputs on PI3K, Akt-pS473, and mTORC1 provided a good fit between simulations and experimental dynamic data for all readouts (Figs 3F and S12). The comparison of the simulations (Fig S12) to our initial model II (Fig S6) with the single stress input on PI3K showed an improved fit for all readouts upon stress and perturbations with wortmannin or MK-2206. In favour of hypothesis (2), model V indeed yielded a lower AIC value (Fig 3E, AIC 1,109) than model III with two stress inputs only (Fig 3C, AIC 1,127). However, as the decrease in AIC was below the threshold of 5%, we decided also to test both hypotheses experimentally.

## mTORC1 activation by stress is independent of Akt and mTORC2, when PI3K is inactive

We first tested the hypothesis that in the absence of PI3K activity, stress-induced Akt-pS473 activates mTORC1 (hypothesis 1, Fig 3C). For this purpose, we inhibited Akt-S473 phosphorylation by Akt1 and Akt2 double knockdown (siAkt1/2) to target the major Akt isoforms expressed in MCF-7 cells (Jordan et al, 2004; Le Page et al, 2006). We analysed mTORC1 activation upon arsenite stress by detecting p70-S6K-pT389 when PI3K was active or inhibited by wortmannin. siAkt1/2 expectedly reduced Akt1 and Akt2 protein levels as well as phosphorylation on T308 and S473 (Figs 4A–C, S13A, and B). When PI3K was active, siAkt1/2 reduced p70-S6K-T389 phosphorylation, indicating that Akt mediates mTORC1 activation by stress (Fig 4A and B). In contrast, when PI3K was inactivated by wortmannin, siAkt1/2 did not further reduce p70-S6K-T389 phosphorylation, although it was still inducible by arsenite stress (Fig 4A and C). This indicated that in the absence of PI3K activity, mTORC1 activation by stress was independent of Akt. Thus, stress-induction of Akt-pS473 does not mediate mTORC1 activation, when PI3K is inactive. We, therefore, rejected hypothesis (1).

To further test this, we inhibited the kinase, which phosphorylates Akt-S473. mTORC2 is widely recognized as the main kinase targeting Akt-S473 (Sarbassov et al, 2005; Jacinto et al, 2006; Manning & Toker, 2017). To disassemble and inhibit mTORC2, we inhibited the specific mTORC2 component rapamycin-insensitive companion of mTOR (rictor) (Sarbassov et al, 2004; Jacinto et al, 2006). Rictor knockdown significantly reduced rictor levels (Fig S13C and D) but did not inhibit p70-S6K-pT389 (Fig 4D–F). This indicates that the stress signal to mTORC1 is not transduced through mTORC2, again supporting the rejection of hypothesis (1).

As expected, rictor knockdown reduced Akt-S473 phosphorylation in PI3K-proficient cells (Fig 4D and E). To our surprise, we found that upon PI3K inhibition, rictor knockdown did not affect Akt-pS473 even though it was induced by arsenite stress (Fig 4D and F). This

indicates that in the absence of PI3K activity, mTORC2 does not mediate Akt-S473 phosphorylation. We further confirmed this finding with Torin1 (Thoreen et al, 2009), an ATP-analogue inhibitor, which targets mTOR in both complexes. Torin1 abolished p70-S6K-T389 phosphorylation (Fig S13E), indicative of mTORC1 inhibition. Akt-S473 phosphorylation was reduced by Torin1 (Fig S13F) to the same level as with wortmannin (Fig S13E), indicating that PI3K and mTORC2 act in the same axis to enhance Akt-pS473. Yet, when PI3K was inactive, Torin1 could not further reduce Akt-pS473, although it remained inducible by arsenite stress (Fig S13G), again suggesting that stress-induction of Akt-pS473 is not mediated by mTORC2 when PI3K is inactive. Which stress-inducible kinase targets Akt-S473 when PI3K is inactive?

## p38/MAPK signaling mediates separate stress inputs to Akt and mTORC1

Next to mTORC2 (Manning & Toker, 2017), at least seven other candidates have been proposed as PDK2s for Akt over the last two decades (DNAPK, ILK, LRRK2, MK2, PKCA, PKCB, and TBK1; Figs 4G and S13H, see S13H for citations). MEDLINE and PubMed Central (PMC) contain >28 million abstracts and >1.7 million full texts, of which more than 50,000 reference the aforementioned seven kinases and/or Akt (Fig 4G). To comprehensively evaluate this literature in an unbiased manner for a kinase that is likely to phosphorylate Akt under stress, we adopted an automated, heuristic strategy. We developed a text mining pipeline, which enabled us to quantitatively evaluate the full text corpus of MEDLINE and PMC, based on numbers of hits. Furthermore, we opted for a linguistic approach that evaluated logical connections between two terms, rather than just counting their co-occurrence.

The text mining pipeline included automated recognition and annotation of genes and proteins in all MEDLINE abstracts and the open access subset of PMC full texts (Fig 4G). Afterwards, we identified sentences from the complete text corpus that fulfilled the following conditions: (i) referenced Akt1, Akt2, Akt3 or any known PDK2 for Akt, (ii) referenced any other gene or protein name and (iii) included a functional relation term like "phosphorylation" or "regulation" that linguistically linked occurrences from (i) and (ii). In total, we obtained 52,413 sentences stating a functional relationship of a protein/gene with Akt or any PDK2. This list was further filtered for the string "stress," to restrict our dataset to stress-related regulatory events (Supplemental Data 6). Based on the number of publications and sentences, in which the hits occurred, we generated a list of targets. The top hit was the MAPK p38 (Fig 4H), which has been proposed to activate the PDK2 candidate MAPKAPK2 under stress (MK2 and MAPK-activated protein kinase 2) (Rouse et al, 1994).

As we aimed to identify the kinase, which in a PI3K-independent manner controls Akt-pS473 under arsenite stress, we next tested whether MK2 is required for Akt-S473 phosphorylation when PI3K is inactive. For this purpose, we used the small compound inhibitor PF3644022 (Mourey et al, 2010) to inhibit MK2 in arsenite-stressed cells (Fig 4I–K). When PI3K was active, PF3644022 did not affect Akt-pS473 (Fig 4I and J). In contrast, when PI3K was inhibited, PF3644022 significantly reduced Akt-S473 phosphorylation, indicating that during stress, MK2 phosphorylates Akt-S473 in a PI3K-independent

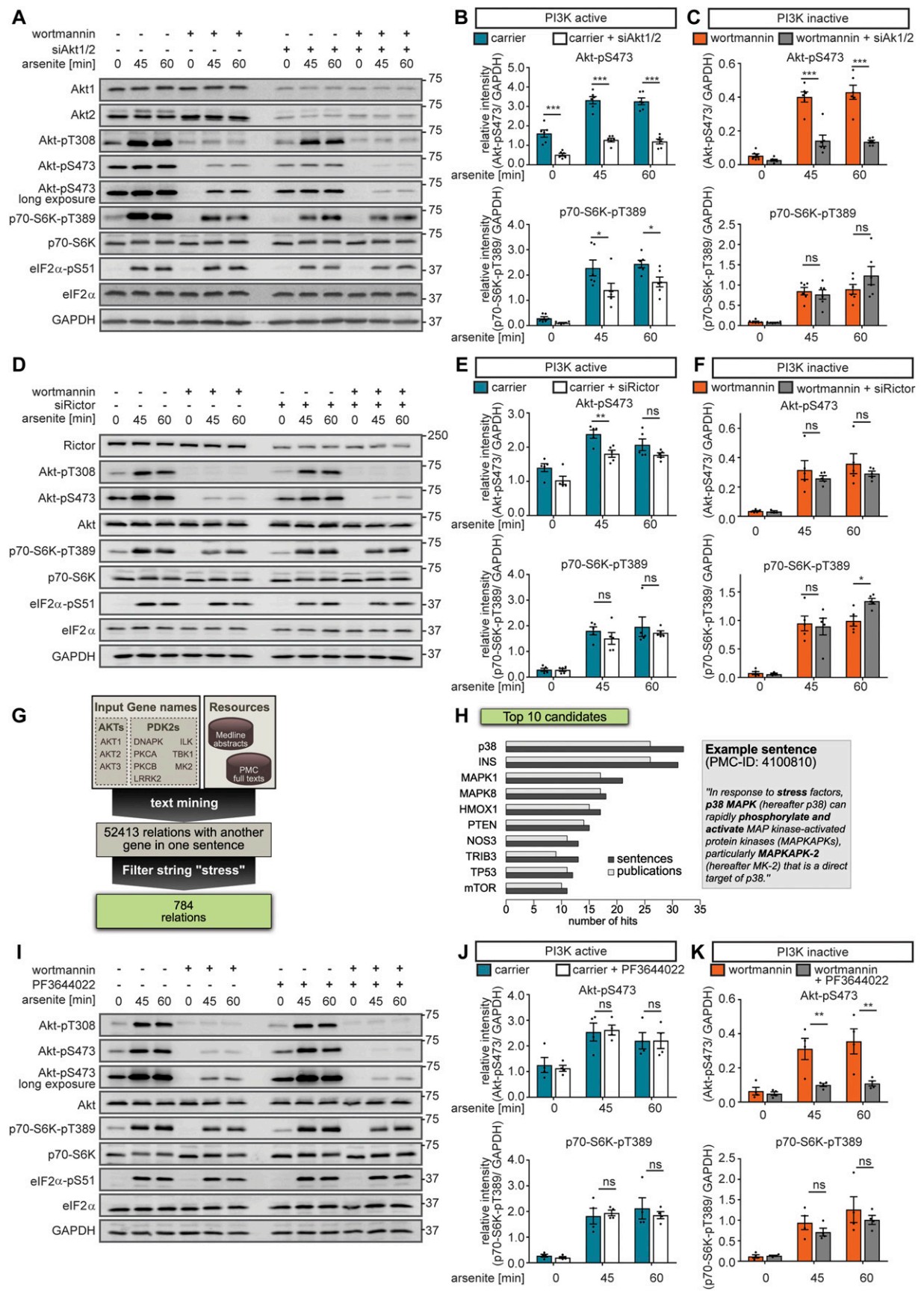

manner (Fig 4I and K). To further confirm our findings, we repeated the same experiment with MK2 siRNA–mediated knockdown, with similar results (Fig S13I–K). We also used a small compound inhibitor of p38 (LY2228820 [Tate et al, 2013; Campbell et al, 2014]), which efficiently inhibited p38 and MK2 as monitored by MK2-T334 phosphorylation (Ben-Levy et al, 1995) (Fig 5A). p38 inhibition also significantly reduced Akt-pS473 (Fig 5A–C), and this effect was more pronounced when PI3K was inhibited. Thus, we conclude that MK2, downstream of p38, is the stress-induced PDK2, which phosphorylates Akt independently of PI3K.

MK2 inhibition did not reduce p70-S6K-pT389 phosphorylation (Figs 4I–K and S13I–K), suggesting that MK2 does not activate mTORC1 under stress. This is in agreement with our finding that Akt does not regulate mTORC1 when PI3K is inactive (Fig 4A and C) and again highlights that stress activation of Akt and mTORC1 are two separate molecular events. In contrast, p38 inhibition did reverse stress-induced p70-S6K-T389 phosphorylation when PI3K was inactive (Fig 5A and C). Thus, we conclude that independently of MK2, p38 mediates stress activation of mTORC1.

Taken together, our findings support hypothesis (2) (Fig 3E, model V), in which three separate stress inputs signal to the mTOR network: (i) PI3K signals through PDK1, Akt-pT308, and the TSC to mTORC1 and through mTORC2 to Akt-pS473; (ii) the p38-MK2 axis signals to Akt-S473; and (iii) mTORC1 is activated by p38 in an MK2-independent manner. Of note, p38 signaling to Akt and mTORC1 is PI3K independent.

### p38 and PI3K promote stress granule formation in a hierarchical manner

Our findings suggest a hierarchy of stress inputs to mTORC1: when highly active, PI3K dominates stress signaling through Akt and the TSC to mTORC1. As PI3K activity declines, p38 takes over to activate mTORC1. This hierarchy can be observed for both bona fide mTORC1 readouts, p70-S6K-pT389 (Fig 5A–C) and 4EBP1-pT37/46 (Figs 5D and

S14A). This hierarchy was also recapitulated by our computational model V, whose simulations showed a much stronger contribution of p38 to mTORC1 activation by stress when PI3K was inactive, as compared with active PI3K (Fig 5E).

To define the threshold of PI3K activity, at which the contribution of p38 to mTORC1 activation by stress becomes apparent, we performed a dose-response experiment. To analyse the effect of p38 inhibition on stress-induced p70-S6K-pT389 at different levels of PI3K activity, we combined the p38 inhibitor LY2228820 with wortmannin concentrations from 10 to 100 nM (Figs 5F and S13L). The effect of p38 inhibition on p70-S6K-pT389 became significant when PI3K activity (calculated based on Akt-pT308) was 40% or lower. Based on this result, the threshold of PI3K activity for p38 to significantly contribute to mTORC1 activation by stress can be estimated to be 40% in MCF-7 cells.

How does p38 activate mTORC1? So far, two mechanisms have been proposed: (i) p38 has been suggested to activate mTORC1 through MK2 (Li et al, 2003; Cully et al, 2010). Yet, in our hands, MK2 inhibition did not affect mTORC1 activity (Figs 4I–K and S13I–K), indicating that mTORC1 activation by p38 was MK2 independent; (ii) p38 has been proposed to directly phosphorylate raptor and activate mTORC1 (Wu et al, 2011), but it is unknown whether this occurs in a PI3K-dependent manner. Hence, in cells with low PI3K activity, p38 might activate mTORC1 either through raptor or via a hitherto unknown mechanism, which requires further investigation in the future.

To test if the hierarchy between PI3K and p38 is a general feature of stress signaling to mTORC1, we investigated this mechanism in four further cell lines, out of which three were of tumor origin (HeLa, cervix carcinoma; CAL51, breast cancer; and LN18, glioblastoma), whereas one was a non-malignant cell line (HEK293T, human embryonic kidney). Surprisingly, we found that all four cell lines exhibited very low p70-S6K levels, as compared with MCF-7, whereas 4E-BP1 levels were similar or higher than in MCF-7 (Fig S14B and C). Thus, we opted for 4E-BP1-T37/46 as the readout to compare the mTORC1 response to p38 inhibition across the five different cell

---

**Figure 4. Stress promotes Akt-S473 phosphorylation via the p38-MK2 axis when PI3K is inactive.**
**(A)** mTORC1 stress activation is not mediated via Akt when PI3K is inactive. MCF-7 cells were serum-starved and treated with arsenite in the presence of carrier (DMSO) or wortmannin (100 nM, PI3K inhibitor) in cells treated with non-targeting scramble siRNA (siControl) or with siRNA-pools targeting Akt1 and Akt2. Akt1, Akt2, Akt-pT308, Akt-pS473, p70-S6K-pT389, and eIF2α-pS51 were monitored by immunoblot. Data represent six biological replicates. **(B)** Quantification of data shown in (A) when PI3K is active. Akt-pS473 and p70-S6K-pT389 levels were compared between siControl and siAkt1/2-treated cells using a two-way ANOVA followed by a Bonferroni multiple comparison test across six biological replicates. Data represent the mean ± SEM. P-values for the Bonferroni multiple comparison tests are shown above the columns. *P ≤ 0.05; ***P ≤ 0.001. **(C)** Quantification of data shown in (A) when PI3K is inactive. Akt-pS473 and p70-S6K-pT389 levels were compared between siControl and siAkt1/2-treated cells in the presence of wortmannin using a two-way ANOVA followed by a Bonferroni multiple comparison tests across six biological replicates. Data represent the mean ± SEM. P-values for the Bonferroni multiple comparison tests are shown above the columns. ns, not significant; ***P ≤ 0.001. **(D)** Akt-pS473 is not mediated by mTORC2 when PI3K is inactive. MCF-7 cells were serum-starved and treated with arsenite in the presence of carrier (DMSO) or wortmannin (100 nM, PI3K inhibitor) in siControl versus siRictor-treated cells. Rictor, Akt-pT308, Akt-pS473 p70-S6K-pT389, and eIF2α-pS51 were monitored by immunoblot. Data represent five biological replicates. **(E)** Quantification of data shown in (D) when PI3K is active. Akt-pS473 and p70-S6K-pT389 were compared between siControl and siRictor using a two-way ANOVA followed by a Bonferroni multiple comparison test across five biological replicates. Data represent the mean ± SEM. The P-values for the Bonferroni multiple comparison tests are shown. **P ≤ 0.01. **(F)** Quantification of data shown in (D) when PI3K is inactive. Akt-pS473 and p70-S6K-pT389 were compared between siControl and siRictor-treated cells in the presence of wortmannin using a two-way ANOVA followed by a Bonferroni multiple comparison test across five biological replicates. Data represent the mean ± SEM. The P-values for the Bonferroni multiple comparison tests are shown. *P ≤ 0.05. **(G)** Workflow of text mining approach. **(H)** List of top 10 identified interaction partners using the text mining approach, including an example sentence (Borodkina et al, 2014). **(I)** MK2 phosphorylates Akt-S473 when PI3K is inactive. MCF-7 cells were serum-starved and treated with arsenite in the presence of carrier (DMSO) or wortmannin (100 nM, PI3K inhibitor). In addition, the cells were treated with carrier (DMSO) or PF3644022 (1 μM, MK2 inhibitor). Akt-pT308, Akt-pS473, p70-S6K-pT389, and eIF2α-pS51 were monitored by immunoblot. Data represent four biological replicates. **(J)** Quantification of data shown in (I) when PI3K is active. Akt-pS473 and p70-S6K-pT389 levels were compared between carrier (DMSO) and PF3644022-treated cells using a two-way ANOVA followed by a Bonferroni multiple comparison test across four biological replicates. Data represent the mean ± SEM. The P-values for the Bonferroni multiple comparison tests are shown. **(K)** Quantification of data shown in (I) when PI3K is inactive. Akt-pS473 and p70-S6K-pT389 levels were compared between wortmannin- and wortmannin + PF3644022–treated cells using a two-way ANOVA followed by a Bonferroni multiple comparison test across four biological replicates. Data represent the mean ± SEM. The P-values for the Bonferroni multiple comparison tests are shown. **P ≤ 0.01. ns, not significant.

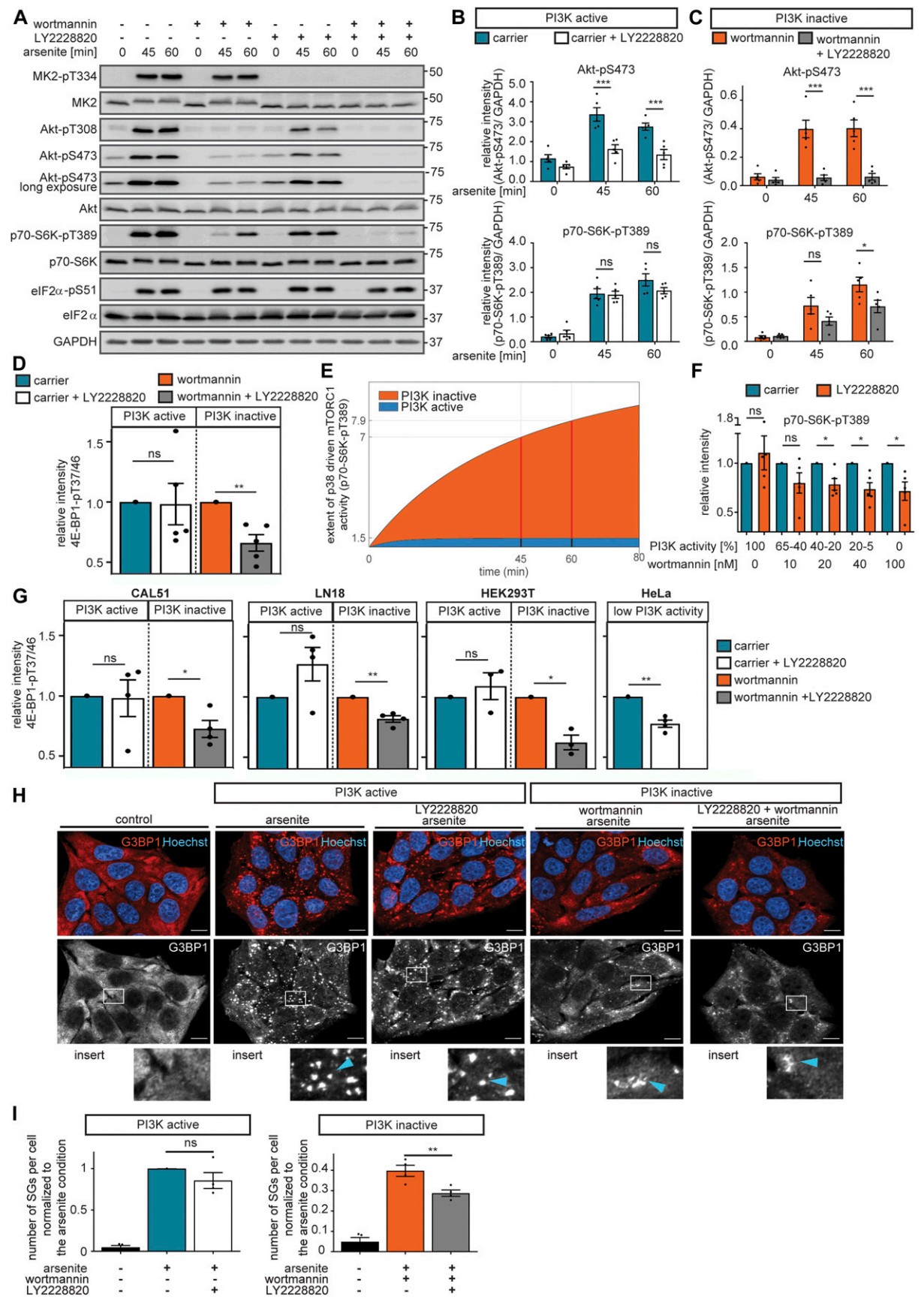

lines (Figs 5G and S14D–G). In cells which exhibited enhanced PI3K activity upon stress (enhanced Akt-pT308 in MCF-7, CAL51, LN18, and HEK293T cells), PI3K inhibition was required to observe an effect of p38 inhibition on 4E-BP1-T37/46 phosphorylation (Figs 5D, G, S14A, and D–F). In HeLa cells, PI3K was not responsive to stress, and here p38 inhibition reduced 4E-BP1-pT37/46 also without additional PI3K inhibition (Figs 5G and S14G). Thus, p38 inhibition directly affects mTORC1 in cells with low PI3K activity; however, in cells with high PI3K, the impact of p38 only becomes observable when PI3K is suppressed. Hence, we conclude that the hierarchy between PI3K- and p38-mediated stress activation of mTORC1 is a general mechanism preserved among different cancer-derived and nonmalignant cell types.

The hierarchy of PI3K and p38 in inducing mTORC1 under stress suggests that the same hierarchy might also exist at the level of stress granule assembly. If so, the effect of p38 inhibition on stress granules would only be observable when PI3K activity is low. To test this, we treated arsenite-stressed cells with the p38 inhibitor LY2228820 without or with PI3K inhibition (wortmannin) and analysed the number of stress granules per cell (Fig 5H and I). Although the p38 inhibitor did not affect stress granule numbers when PI3K was active, p38 inhibition did reduce the amount of stress granules in cells with inactive PI3K. Thus, we identify p38 as a novel regulator of stress granules.

We conclude that the two pro-stress-granule-kinases PI3K and p38 drive stress granule assembly in a hierarchical manner, as the importance of p38 for stress granule formation becomes apparent as PI3K activity declines.

## Discussion

The present work is the first attempt to model the full mTOR network, including both mTOR complexes, to identify activating stress inputs and their co-dependency. Furthermore, we generated and applied an automated, heuristic text mining pipeline, which allowed us to quantitatively evaluate the full text corpus of MEDLINE and PMC to comprehensively evaluate connections between signaling components under stress. Based on model-guided experimentation, we established that PI3K and p38 activate mTORC1 in a hierarchical manner (Fig 6A). Only the understanding of this hierarchy at the level of signaling enabled the discovery of p38 as a pro-stress-granule-kinase.

Although inhibitory or activating roles of mTORC1 in stress granule assembly have been postulated (Fournier et al, 2013; Hofmann et al, 2012; Panas et al, 2016; Sfakianos et al, 2018), we find here that mTORC1 acts as a pro-stress-granule-kinase. How does this function relate to earlier findings that stress granules inhibit mTORC1 (Thedieck et al, 2013; Wippich et al, 2013)? Although stress signaling through PI3K and p38 to mTORC1 enhances stress granule formation, stress granules in turn inhibit mTORC1 activity, thereby likely restricting their own assembly. This dual mechanism implies a negative feedback loop that might contribute to the tight balance of stress granule formation and clearance, to allow cell survival under stress.

Stress granules are induced by a variety of stresses, including heavy metals, oxidative, ER (endoplasmic reticulum), nutritional, heat, and osmotic stress; UV and γ irradiation; and chemotherapeutic agents (Anderson et al, 2015). Do PI3K and p38 respond to similar stresses? Indeed, PI3K is activated by heavy metals (Carpenter and Jiang, 2013), γ irradiation (Datta et al, 2014), heat (Thompson et al, 2016), and oxidative stress (Kosmidou et al, 2001). Also, p38 responds to many stressors known to enhance stress granules, including ER stress, oxidative stress, nutritional stress, and DNA damage (Bonney, 2017). Thus, PI3K and p38 may be universal pro-stress-granule-kinases.

So far, mechanisms which suppress translation under stress have been perceived as the main triggers of stress granule assembly. Most

---

**Figure 5. p38 promotes mTORC1 activation and stress granule formation when PI3K is inactive.**
**(A)** p38 mediates mTORC1 activation when PI3K is inactive. MCF-7 cells were serum-starved and treated with arsenite in the presence of carrier (DMSO) or wortmannin (100 nM, PI3K inhibitor). In addition, the cells were treated with carrier (DMSO) versus LY2228820 (1 μM, p38 inhibitor). MK2-pT334, Akt-pT308, Akt-pS473, p70-S6K-pT389, and eIF2α-pS51 were monitored by immunoblot. Data represent five biological replicates. **(B)** Quantification of data shown in (A) when PI3K is active. Akt-pS473 and p70-S6K-pT389 were compared between carrier (DMSO) and LY2228820-treated cells using a two-way ANOVA followed by a Bonferroni multiple comparison test across five biological replicates. Data represent the mean ± SEM. The *P*-values for the Bonferroni multiple comparison tests are shown. ***$P$ ≤ 0.001. **(C)** Quantification of data shown in (A) when PI3K is inactive. Akt-pS473 and p70-S6K-pT389 were compared between wortmannin- and wortmannin + LY2228820–treated cells using a two-way ANOVA followed by a Bonferroni multiple comparison test across five biological replicates. Data represent the mean ± SEM. The *P*-values for the Bonferroni multiple comparison tests are shown. *$P$ ≤ 0.05; ***$P$ ≤ 0.001. **(D)** p38 drives mTORC1 activity when PI3K is inactive. Quantification of data shown in Fig S14A. 4E-BP1-pT37/46 relative intensity was normalized separately for conditions without or with wortmannin. Significance of 4E-BP1-pT37/46 inhibition by LY2228820 was tested using a two-tailed *t* test across five biological replicates. Data represent the mean ± SEM. *$P$ ≤ 0.05. **(E)** Prediction on the extent of mTORC1 inhibition upon LY2228820 treatment when PI3K is active or inactive. Prediction was performed with model V. The red lines depict the time points measured experimentally (Fig 5A–C). **(F)** When PI3K activity declines, p38 drives mTORC1 activity. Quantification of data shown in Fig S13L. MCF-7 cells were serum-starved and treated with arsenite for 60 min in the presence of different concentrations of wortmannin (as indicated, PI3K inhibitor) in carrier (DMSO) versus LY2228820 (1 μM, p38 inhibitor)-treated cells. p70-S6K-pT389 relative intensity was normalized separately for each wortmannin concentration. Significance of p70-S6K-pT389 inhibition by LY2228820 was tested using a two-tailed *t* test across five biological replicates. Data represent the mean ± SEM. *$P$ ≤ 0.05. **(G)** p38 drives mTORC1 activity in several cell lines, as PI3K activity declines. Quantification of data shown in Fig S14D–G. MCF-7, CAL51, LN18, HEK293T, and HeLa cells were serum-starved and exposed to arsenite for 60 min in combination with wortmannin (100 nM, PI3K inhibitor) and/or LY2228820 (1 mM, p38 inhibitor). Data represent 3–4 biological replicates (see Fig S14D–G). 4E-BP1-pT37/46 relative intensity was normalized separately for conditions without or with wortmannin. Significance of 4E-BP1-pT37/46 inhibition by LY2228820 was tested using a two-tailed *t* test across three biological replicates. Data represent the mean ± SEM. *$P$ ≤ 0.05; **$P$ ≤ 0.01. **(H)** Stress granule numbers upon PI3K and p38 inhibition. MCF-7 cells were serum-starved and treated with arsenite for 30 min in the presence of carrier (DMSO), wortmannin (100 nM, PI3K inhibitor), LY2228820 (1 μM, p38 inhibitor), or wortmannin + LY2228820. Stress granules were visualized by immunofluorescence staining of G3BP1. Nuclei were visualized with Hoechst 33342. Data represent four biological replicates. White square indicates region of insert and blue arrow highlights stress granules; scale bar 10 μm. **(I)** Quantification of data shown in (H). The number of stress granules (SGs) per cell (normalized to the arsenite condition) across four biological replicates. Stress granule formation between carrier and LY2228820 as well as wortmannin- and wortmannin + LY2228820–treated cells was compared using a two-tailed *t* test across four biological replicates. Data represent the mean ± SEM. *$P$ ≤ 0.01. ns, not significant.

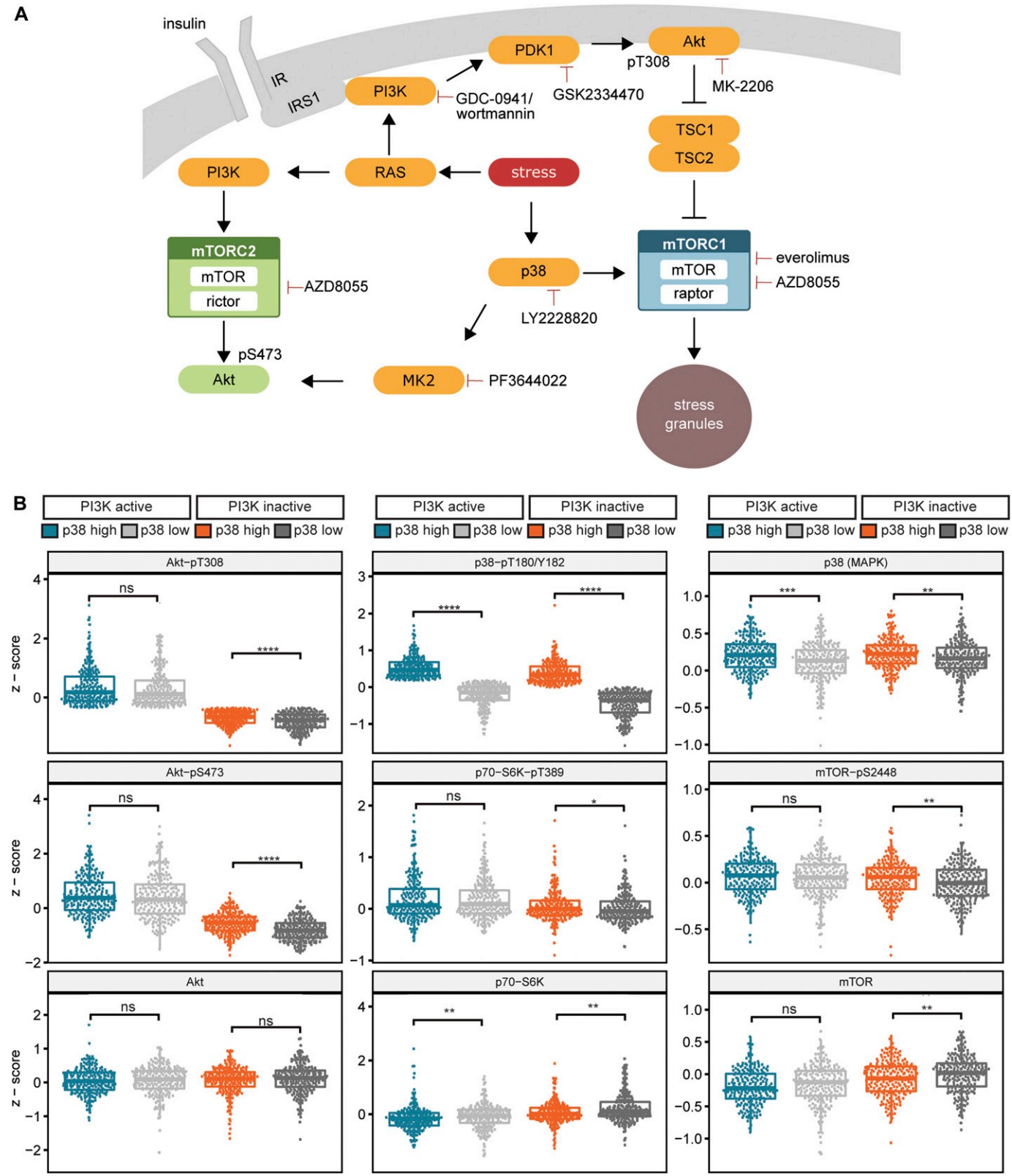

**Figure 6. PI3K, p38, and MK2 transduce stress signals to the mTOR network.**
**(A)** Scheme summarizing the stress response of the mTOR network. Three distinct cues mediate stress signals to the mTOR network. (1) Stress activates the RAS-PI3K-PDK1-Akt axis, which in turn activates mTORC1. (2) The p38-MK2 axis signals to Akt-pS473. This stress input does not activate mTORC1. (3) p38 activates mTORC1, independently of PI3K. Stress signaling to mTORC1 is required for stress granule assembly. The inhibitors used in this study are depicted. **(B)** p38 and mTORC1 activity specifically correlate in

prominently, eIF2α kinases are considered as major stress granule triggers (Anderson et al, 2015), as eIF2α kinases inhibit overall cap-dependent translation by preventing the assembly of the ternary complex (Holcik, 2015). In contrast to eIF2α kinases, the pro-stress-granule-kinases PI3K and p38 identified here, as well as their downstream target mTORC1, are perceived as translation enhancers (Gonskikh & Polacek, 2017; Robichaud & Sonenberg, 2017; Saxton & Sabatini, 2017). So far, the oncogenic effects of PI3K, p38, and mTORC1 have been mainly assigned to the fact that they enhance metabolism, thereby promoting cell growth. Our results suggest that through stress granule formation, PI3K, p38, and mTORC1 also promote cell survival, adding to their oncogenic capacity. Hence, stress granule inhibition might be a valuable marker to monitor the effectiveness of therapies targeting PI3K, p38, and mTORC1. Of note, stress granule disruption by compounds has been hitherto difficult (Hu et al, 2017; Timalsina et al, 2018). The understanding of signaling cues mediating stress granule assembly provides us new tools, such as PI3K and p38 inhibitors, to target stress granules for research and therapies.

Also, the hierarchy of PI3K and p38 signaling to mTORC1 and stress granules, established in the present study is of high relevance for cancer treatment, as ongoing clinical trials combine the p38 inhibitor LY2228820 with genotoxic therapies (https://clinicaltrials.gov/ct2/show/NCT02364206 and https://clinicaltrials.gov/ct2/show/NCT01663857), known to induce stress granules (Moeller et al, 2004). Our findings suggest that this is a promising strategy as LY2228820 inhibits stress induction of mTORC1 and stress granule formation, and this mechanism might be predominant in cancers with moderate or low PI3K activity. To investigate this in patient tissues, we analysed protein array data of the invasive breast cancer dataset from The Cancer Genome Atlas (TCGA). Indeed, we found in tumors with low PI3K activity that mTORC1 readouts (p70-S6K-pT389 and mTOR-pS2448) as well as Akt-S473 phosphorylation correlated with p38 activity (p38-pT180/Y182 [Corre et al, 2017]) (Fig 6B). In contrast, when PI3K activity was high, phosphorylation of Akt-S473 and mTORC1 substrates did not correlate with p38 phosphorylation. This suggests that p38 specifically activates Akt and mTORC1 in tumors with low PI3K activity and that such patients may exhibit a better response rate to p38 inhibitors. Thus, we advocate cautious monitoring of PI3K readouts in clinical trials to predict treatment outcome for p38 inhibitors.

# Materials and Methods

## Cell culture and cell treatments

Experiments were performed in MCF-7 cells overexpressing LC3-GFP, a gift from Joern Dengjel. CAL51 (Cat. No. ACC 302) cells were obtained from the Leibniz Institute DSMZ–German Collection of Cell Cultures and Microorganisms. LN18 (Cat. No. CRL-2610) cells were obtained from ATCC. HEK293T cells were obtained from the clinical department of Freiburg University (Thien et al, 2015). HeLa, MCF-7, and HEK293T cells were validated by DSMZ for their origin using short tandem repeat analysis. The cells were cultured in DMEM (Cat. No. P04-03600; PAN Biotech) supplemented with 10% FBS (Cat. No. 10270-106; Gibco) and 3 mM L-glutamine (Cat. No. 25030-024; Gibco) at 37$\underline{o}$C and 7.5% $CO_2$. All cells were tested every three months for mycoplasma contamination by performing a PCR on the cell supernatant. 16 h before arsenite treatment (Cat. No. 3500-1L-R; Fluka analytic), the cells were washed twice with PBS (Cat. No. P04–36500; PAN) and afterwards cultivated in DMEM supplemented with 3 mM L-glutamine at 37$\underline{o}$C and 7.5% $CO_2$ without FBS (serum starvation). 500 $\mu$M arsenite was added to the starvation medium for the indicated durations. For inhibitor treatment, inhibitors were added 30 min before the arsenite treatment. Control cells were treated with the corresponding carrier (DMSO) (Cat. No. D2650; Sigma-Aldrich) at the same final concentration.

siRNA knockdown was induced as follows: to induce IRS1 knockdown, 5 nM siRNA single sequence were used for transfection and incubated for 3 days (siIRS1 #1, target sequence UCAAAGAGGU-CUGGCAAGU; siIRS1 #2 target sequence GAACCUGAUUGGUAUCUAC; Dharmacon, Cat. No. LU-003015-00). ON-TARGETplus Human SMARTpool (Dharmacon) was used to induce knockdown of the following proteins: for Akt1 (Cat. No. L-003000-00) and Akt2 (Cat. No. L-003001-00) knockdown, 10 nM siRNA was used for two days. To knockdown rictor (Cat. No. L-016984-00), 20 nM siRNA was used for 4 days. To knockdown MK2 (Cat. No. L-003516-00), 20 nM siRNA was used for 3 days. As a control, non-targeting scramble siRNA (siControl, Cat. No. D-001810-10) was used at the same concentration as the target siRNA. siRNA transfections were performed using Lipofectamine 3000 (Cat. No. L3000008; Invitrogen) according to the manufacturer's protocol.

## Antibodies and reagents

The following antibodies were purchased from Cell Signaling Technology: p70-S6K-pT389 (Cat. No. 9206), p70-S6K (Cat. No. 9202), PRAS40-pS183 (Cat. No. 5936), PRAS40-pT246 (Cat. No. 2997), PRAS40 (Cat. No. 2610), 4E-BP1-pT37/46 (Cat. No. 2855), 4E-BP1 (Cat. No. 9644), Akt-pT308 (Cat. No. 4056), Akt-pS473 (Cat. No. 9271), Akt (Cat. No. 9272), Akt1 (Cat. No. 2938), Akt2 (Cat. No. 3063), TSC2-pT1462 (Cat. No. 3617), TSC2 (Cat. No. 4308), IRS1-pS636/639 (Cat. No. 2388), IRS1 (Cat. No. 2382), eIF2α-pS51 (Cat. No. 9721), eIF2α (Cat. No. 9722), MK2-pT334 (Cat. No. 3007), MK2 (Cat. No. 3042), and RAF (Cat. No. 2662). GAPDH antibody was bought from Abcam (Cat. No. ab8245), rictor antibody from Bethyl Laboratories (Cat. No. A300-459A), and S6K-pT229 antibody from GeneTex (Cat. No. GTX25231). Antibodies from Cell Signaling Technology and GeneTex were used at a final dilution of 1:1,000 in TBST (Tris-buffered saline-Tween) buffer (138 mM sodium chloride, 2.7 mM potassium chloride, 66 mM Tris (pH 7.4), and 0.1%

breast cancer tissues with low PI3K activity. Analysis of RPPA from breast cancer patients obtained from TCPA. Patients were initially divided by the median of Akt-T308 phosphorylation into two groups of (i) PI3K active and (ii) PI3K inactive. The PI3K active and PI3K inactive groups were further divided by the median of the p38-pT180/Y182 phosphorylation into high p38 and low p38 activity groups. This separated the patients into four groups, namely, (1) PI3K active and p38 high, (2) PI3K active and p38 low, (3) PI3K inactive and p38 high, and (4) PI3K inactive and p38 low. Data represent single measurements. *$P \leq 0.05$; **$P \leq 0.01$; ***$P \leq 0.001$; ****$P \leq 0.0001$. ns, not significant.

Tween-20) with 5% BSA (Cat. No. 8076; Carl Roth). The anti-GAPDH monoclonal antibody was used at a final dilution of 1:10,000 in TBST buffer with 5% BSA. All primary antibodies were used in 5% BSA in TBST supplemented with 0.1% NaN$_3$. Horseradish peroxidase-conjugated goat anti-mouse (Cat. No. 31430) and goat anti-rabbit IgG (Cat. No. 31460) were ordered from Thermo Scientific Pierce and were used at a final dilution of 1:8,000 in TBST buffer with 5% BSA. For immunofluorescence experiments, G3BP1 (Cat. No. sc-81940; Santa Cruz) was used in a dilution of 1:200 as primary antibody in 0.3% FBS in PBS. Goat anti-mouse IgG Alexa Fluor 568 conjugate (Cat. No. A-11031; Thermo Fisher Scientific) was used as secondary antibody for immunofluorescence experiments. For the RAS activity assay, RAS (Cat. No. 05-516; Millipore) and RAF (Cat. No. 2662; Cell Signaling Technology) antibodies were used at a dilution of 0.5 μg/ml and 1:1,000 respectively. Secondary antibodies were purchased from Li-Cor Bioscience and used at a final dilution of 1:10,000: IRDye 680 Goat anti-Mouse IgG (Cat. No. 926-32220), IRDye 680 Goat anti-Rabbit IgG (Cat. No. 926-32210), IRDye 800 Goat anti-Mouse IgG (Cat. No. 926-32220), and IRDye 800 Goat anti-Rabbit IgG (Cat. No. 926-32211).

The inhibitors used in this study were everolimus (Cat. No. S1120; Selleck), wortmannin (Cat. No. 1232; Tocris), PF3644022 (Cat. No. PZ0188-5MG; Sigma-Aldrich), Torin1 (Cat. No. 1833; Axon-Medchem), AZD8055 (Cat. No. 1561; Axon Medchem), LY2228820 (Cat. No. 1895; Axon Medchem), GDC-0941 (Cat. No. 1377; Axon Medchem), GSK2334470 (Cat. No. 4143/10; Tocris), and MK-2206 (Cat. No. S1078; Selleckchem).

## Cell lysis and immunoblotting

For lysis, the cells were washed twice with PBS and lysed with radio immunoprecipitation assay buffer (1% IGEPAL CA-630, 0.1% SDS, and 0.5% sodium deoxycholate in PBS) supplemented with cOmplete Protease Inhibitor Cocktail (Cat. No. 11836145001; Sigma-Aldrich), Phosphatase Inhibitor Cocktail 2 (Cat. No. P5726; Sigma-Aldrich), and Cocktail 3 (Cat. No. P0044; Sigma-Aldrich). Protein concentration was measured using Bio-Rad Protein Assay Dye Reagent Concentrate (Cat. No. 500-0006; Bio-Rad) and adjusted to the lowest value. Cell lysates were mixed with sample buffer (10% glycerol, 1% β-mercaptoethanol, 1.7% SDS, 62.5 mM TRIS base (pH 6.8), and bromophenol blue) and heated for 5 min at 95°C. Proteins were then fractionated by size using SDS–PAGE. Acrylamide percentage ranged from 8 to 14%, depending on the size of the proteins to be detected. For separation, 90 to 180 V was applied and a Mini-PROTEAN Tetra Vertical Electrophoresis Cell system (Cat. No. 1658029FC; Bio-Rad) with running buffer (0.2 M glycine, 25 mM Tris base, and 0.1% SDS) was used. For protein membrane transfer, we used polyvinylidene difluoride membranes (Cat. No. IPVH00010; Merck) and the PROTEAN Tetra Vertical Electrophoresis Cell system with blotting buffer (0.1 M glycine, 50 mM TRIS base, 0.01% SDS (pH 8.3), and 10% methanol) at 45 V for 1 h and 50 min. The membranes were blocked in 5% BSA in TBST. Primary antibodies were incubated overnight at 4°C, and afterwards, the membranes were washed and incubated for at least 2 h with the HRP-coupled secondary antibody (goat–anti-mouse and goat–anti-rabbit). For detection, Pierce ECL Western blotting substrate (Cat. No. 32209; Thermo Fisher Scientific) or SuperSignal West FEMTO (Cat. No. 34095; Thermo Fisher

Scientific) was used to detect chemiluminescence using an LAS-4000 mini camera system (GE Healthcare). Raw images taken with the LAS-4000 mini system were exported as red green blue (RGB) colour TIFF files using ImageJ and further processed with Adobe Photoshop version CC2018. Quantification of raw image files was performed using ImageQuant TL version 8.1 (GE Healthcare). Background subtraction was performed using the rolling ball method, implemented in ImageQuant TL software, with a defined radius of 200 for all images. The pixel value of a given signal was normalized to the average pixel intensity of all signals belonging to the respective readout in a given experiment. Afterwards, the signals were normalized to the loading control (GAPDH). The GraphPad Prism 8 software was used for statistical analysis. The number of biological replicates for each experiment is indicated in the figure legends, and the single experimental values are depicted as dots in the graphical representations. A one-way or two-way ANOVA followed by a Bonferroni multiple comparison test or a two-tailed $t$ test was applied as indicated in the figure legends to compare the signal intensities among different conditions across the experiments.

## Immunofluorescence

For all immunofluorescence experiments, the cells were washed with PBS and fixed with 4% paraformaldehyde (Cat. No. 1.04005.1000; Merck) in PBS for 5 min at room temperature. After washing the cells three times with PBS, permeabilization was performed with 0.1% Triton X-100 (Cat. No. 93443; Sigma-Aldrich) for 5 min at room temperature. The cells were washed in PBS and blocked with 0.3% FBS in PBS for 30 min at room temperature. Primary antibodies were diluted in 0.3% FBS in PBS and incubated overnight at 4°C in a humid chamber. Next, the cells were washed with PBS and incubated with secondary antibody (Alexa-568) and Hoechst 33342 (end concentration 1 mg/ml; Cat. No. H3570; Invitrogen) diluted in 0.3% FBS in PBS for 30 min in a humid dark chamber at room temperature. Slides (Cat. No. 4951PLUS4; Thermo Fisher Scientific) were washed with PBS and water, and mounted with Mowiol 4–88 solution (Cat. No. 07131; Carl Roth), which was prepared according to the manufacturer's instructions, including sodium deoxycholate (Cat. No. D27802; Sigma-Aldrich) supplemented with 10% n-propyl-gallate (NPG; Cat. No. 8.205.990.100; VWR International). The slides were analysed using fluorescence microscopy. Z stack images (8 slides, 0.5 μm) were taken with an AxioObserver Z1 compound microscope (Carl Zeiss) with an Apotome (5 pictures per slide), 63× objective and an AxioCam MRm3 CCD camera (Carl Zeiss). For quantitative analysis, maximum intensity projections were generated from three to four representative fields of view, which were captured for each condition with identical exposure times and the same magnification. For presentation in figures, maximum intensity projections were exported as TIFF with no compression using Zen2012 blue edition software (Zeiss), and brightness or contrast were adjusted for better visibility. Brightness or contrast adjustment was not performed before quantification, and thus did not influence image quantification. For quantification of stress granules, G3BP1-positive foci and number of cells were manually counted over four representative fields of view for each biological replicate. The relative number of stress granules per cell

was calculated as the ratio between G3BP1-positive foci and number of cells for each field of view. The average of all representative fields from each condition was normalized to the arsenite condition, which was set to 1. Statistics were performed with GraphPad Prism 8 by comparing the relative amounts of stress granules per cell over all biological replicates between two conditions using a two-tailed $t$ test.

### RAS assay

The RAS activity was studied on the basis of RAS–RAF interaction using RBD agarose beads (Cat. No. 17-218; Millipore) according to the manufacturer's specifications. The cells were treated with arsenite for the specified duration and were subsequently lysed in magnesium lysis buffer (25 mM Hepes (pH 7.5), 150 nM NaCL, 1% IGEPAL CA-630, 10 mM $MgCl_2$, and 1 mM EDTA) for 30 min on ice. Debris was removed by centrifugation at 15,871 $g$ for 15 min at 4°C, and cleared lysates were subsequently used for protein measurement according to the manufacturer's instructions using Bradford reagent (Cat. No. 1856209; Pierce). Equal amounts of all protein lysates were subsequently incubated with RBD agarose beads for 45 min at 4°C and thereafter washed three times with magnesium lysis buffer by centrifugation. Agarose beads were then resuspended in 40 $\mu$l of 2× sample buffer, boiled for 5 min, and beads were collected by brief centrifugation. 20 $\mu$l per sample were resolved on SDS–PAGE with appropriate acrylamide percentage and transferred onto nitrocellulose membranes (Cat. No. 10600002, Amersham; GE Healthcare Life Sciences). The membranes were blocked with 5% non-fat dry milk in TBST for 1 h at room temperature and thereafter incubated overnight at 4°C with either anti-Ras or anti-GST primary antibodies diluted in 5% non-fat dry milk with TBST. A washing step with TBST followed, and the membranes were subsequently probed with secondary antibodies conjugated to either horseradish peroxidase (anti-rabbit/mouse HRP-linked IgG antibody) or to a fluorophore for 1 h at room temperature. The membranes were washed again as described above, and chemiluminescence was detected using GE Healthcare ECL Western blotting detection reagents (Cat. No. RPM2106; Ammersham ECL) and FluorChem M Scanner, whereas fluorescence was detected using the Li-Cor Odyssey Infrared System.

### Mathematical model

The final ordinary differential equation–based model including a stress-related input on PI3K, on Akt-pS473, and on mTORC1 (model V) comprises 25 species and 91 parameters. In general, the model describes the activity state (e.g., phosphorylation) of the following species: IR-$\beta$, IRS1, PI3K, PDK1, Akt, PRAS40, TSC2, p70-S6K, and 4E-BP1. For the sake of simplicity, different compartments and cell volume were neglected. All species activations and deactivations were simulated using mass action kinetics. Inhibitions were simulated to be concentration dependent, that is, the efficiency of inhibition correlates with the concentration of the inhibiting species relative to its total concentration (including active and inactive variants based on phosphorylation states). If the concentration of the inhibiting species variant (e.g., TSC2) equals that of the total concentration of all species variants (e.g., TSC2 + TSC2_pT1462), then that species variant is fully inhibiting its downstream target. A

Boolean-like inhibition scheme was applied for inhibitory inputs to the model, for example, wortmannin or MK-2206, where only in the presence of these components, their respective targets get fully (PI3K) or partially (Akt) inhibited.

### Parameter settings

The model was parameterised on the following experimental datasets: arsenite (Fig S1C), arsenite + wortmannin (Fig S3A), and arsenite + MK-2206 (Fig S3C). The average relative intensities and standard deviations were determined for the following observable species: Akt_pT308, Akt_pS473, TSC2_pT1462, PRAS40_pS183, PRAS40_pT246, p70_S6K_pT389, p70_S6K_pT229, 4EBP1_T37/46, and IRS1_pS636 (Supplemental Data 7). In the case of Akt, PRAS40 and p70_S6K immunoblot data were linked to the total amount of phosphorylation of the respective species (e.g., Akt_pS473 phosphorylation was linked to the sum of Akt_pS473 and Akt_pT308_pS473). As at time point zero of arsenite treatment the initial species concentrations vary across datasets, we included condition-specific initial concentration parameters (arsenite, arsenite + wortmannin, and arsenite + MK-2206). To keep the number of parameters as low as possible, we integrated several assumptions: because mTORC1 and mTORC2 activities were not measured directly, but rather through the phosphorylation of their respective downstream targets, mTORC1 and mTORC2 were modelled implicitly via their downstream effects (e.g., inhibition of mTORC1 by TSC2 was realised by adding TSC2 inhibition to the phosphorylation of all mTORC1 readouts [Figs S2B, S5, S7, S9, and S11]). We used PRAS40_pS183, p70_S6K_pT389, and 4EBP1_T37/46 as mTORC1 activity readouts and Akt_pS473 as mTORC2 activity readout. Because MCF-7 cells were starved overnight against growth factors in all experiments, activation via insulin was not included. The value for the inhibition by MK-2206 and wortmannin was derived from the corresponding readouts (Fig S2C–E). To reduce the number of model parameters and as because of serum starvation, the mTOR network activity is at baseline, we set the initial concentration of all double-phosphorylated species (Akt_pT308_pS473, PRAS40_pT246_pS183, and p70_S6K_pT229_pT389) to zero. Because the effectiveness of a second phosphorylation for AGC kinases such as Akt and p70_S6K is enhanced by its first phosphorylation (Pearce et al, 2010), we applied a factor of two to the second phosphorylation step relative to the parameter responsible for the first phosphorylation. The exact value of the second step phosphorylation enhancement does not affect the overall fitting quality and the outcome of the predictions. We finally assumed that dephosphorylation of the same phosphosite for the species Akt, PRAS40, and p70_S6K is equal, regardless of the species' current phosphorylation state (e.g., the parameter for the dephosphorylation of Akt_pS473 to Akt possesses the same value as the parameter for the dephosphorylation of Akt_pT308_pS473 to Akt_pT308). A summary of all simulated species, inputs, and parameters is provided as supplemental information (Figs S15 and S16 and Tables S2 and S3).

### Input search

The MATLAB toolbox data2dynamics (Raue et al, 2015), release version 170117, was used for model setup, stress input search, and parameter estimation. In addition to the available datasets that are

presented in the data figures (Figs S1C, S3A, and C and Supplemental Data 7), artificial datasets without any stress input were used. The additional data comprise constant values represented by the initial values of the corresponding datasets (e.g., time point zero values of the arsenite dataset [Fig S1C and D] were used for all values of the artificial dataset without arsenite). This is necessary to prevent artificial dynamic behaviour of the model. The feasible range for parameter estimation of initial concentration parameters was set to $10^{-5}$ and $10^1$, except for Akt_pT308 under wortmannin treatment, which was expressed as a fraction of the arsenite dataset-specific initial concentration parameter, because of the inhibitory effect of wortmannin on PI3K. The fit quality of each concurrent model version was assessed by scanning the solution space with 500 repetitions of Latin hypercube sampling (LHS) and gradient descent (Raue et al, 2013). The fit quality was determined by computing the Akaike information criteria (AIC). Also an AIC version including a correction for small sample sizes (AICc) and the Bayesian information criterion (BIC) were computed to assess and compare fit quality (Tables 1 and S1). Because of the semi-quantitative nature of the immunoblot data, initial parameter ranges for all parameters were set to $10^{-5}$ and $10^3$ if not stated otherwise (Figs S15 and S16 and Tables S2 and S3). All mass action–related parameter ranges were set to $10^{-5}$ and $10^1$, except for parameters related to PI3K activation. The corresponding initial parameter ranges were set to $10^{-5}$ and $10^{-2}$ to prevent an unrealistic step-wise activation of the unobservable PI3K species upon stress. The input search performed to find the second stress input was performed using the whole parameter space described above.

### Parameter estimation and identifiability analysis

For the final model (Fig S11 and Tables S2 and S3), an additional parameter estimation was performed by an iterative approach of profile likelihood estimation (Raue et al, 2009) (PLE) and LHS500 (500 different starting points for LHS) runs, starting with the parameters associated with the initial concentration species, followed by fixing the remainder of free parameters. To narrow down the solution space of free parameters, each subsequent parameter identification iteration was based on fixed parameters of the last rounds. Moreover, to reduce the parameter space for the model with three stress inputs (model V), we created an artificial dataset under the assumption that remaining levels of Akt-pS473, p70-S6K-pT389, 4E-BP1-pT37/46, and PRAS40-pS183 after wortmannin inhibition are due to an additional input. The iterative process for parameter estimation was initialized with an LHS500 run to find parameter values that best resembled our data. Next, a PLE run was performed, resulting in a first set of identifiable parameters that were consequently fixed. If PLE reported no identifiable parameter, another LHS500 analysis was performed. The solutions of the LHS500 run were analysed for the best portion of fitting solutions by analysing the goodness of fit of all 500 solutions with k-means clustering. Specifically, two clusters were sought and only solutions corresponding to the cluster with the better mean $\chi^2$ fitting value were kept for further analysis (see Supplemental Data 8 for details). The remaining parameter distributions were analysed per parameter for showing unimodal distributions, that is, ending nearly always in the same value range regardless of the starting point for

parameter optimisation (Supplemental Data 1). Parameters with unimodal value distributions were fixed by identifying the maximum value of a nonparametric kernel-smoothing distribution over each parameter-specific unimodal distribution. After this step, another parameter iteration round was started with a new PLE analysis until no remaining free parameters were left.

### Text mining

To automatically mine publications that relate Akt or PDK2s with other interactors, MEDLINE and the open access subset of PMC were linguistically preprocessed. Required text analysis before actual text mining (such as part of speech tagging) includes segmentation of text into basic units (sentences and words), acronym recognition, and shallow grammatical analysis, among others, and was conducted by using tools from the JCore repository (Hahn et al, 2016). Eligible texts for text mining included more than 28 million abstracts from MEDLINE as of January 2018 and about 1.7 million full texts from PMC (open access only) as of October 2017. Next, a semantic analysis was carried out to identify the mention of genes and interactions between them. Gene identification and mapping into the NCBI Gene database was performed using GeNo (Wermter et al, 2009), whereas interactions were identified by using the tool BioSem (Bui & Sloot, 2012). Next, the sentences in which BioSem identified interactions were filtered for the occurrence of the term "stress" and scanned for references of Akt or PDK2s as interaction member. Finally, results were manually curated to obtain the final result list of literature mentioning interaction events including Akt or PDK2s (Supplemental Data 6).

### TCGA data analysis

Harmonized HT-Seq counts of TCGA breast cancer patients were downloaded from GDC (https://gdc.cancer.gov/) using TCGAbiolinks (Colaprico et al, 2015), and only patients with the identifier "primary solid tumor" were retained. Level-4–normalized reverse phase protein arrays (RPPA) of TCGA breast cancer patients were downloaded from The Cancer Proteome Atlas (TCPA) (http://tcpaportal.org/tcpa). The patient dataset was reduced to the overlap between both RPPA and RNA-seq datasets, and this dataset was used for all further downstream analysis (n = 869). Patients were initially divided by the median of Akt_pT308 phosphorylation into two groups of high and low phosphorylation. In addition, the high or low Akt_pT308 group was further divided by the median of the p38_pT180Y182 phosphorylation into high and low groups. This grouped the patients into four groups. Wilcoxon signed-rank test was applied for group comparisons of Akt, Akt_pT308, Akt_pS473, p38MAPK, p38_pT180Y182, p70-S6K, p70-S6K_pT389, mTOR, and mTOR_pS2448 measurements. All analyses were run in R, version 3.4.4 (https://cran.r-project.org/), with the Bioconductor version 3.6 (https://bioconductor.org/). All graphical representations were generated using ggplot2, ggpubr, and ggbeeswarm.

### Data availability

The five models generated as a part of this study are provided as Supplementary Data. Supplemental Data 1: Model without stress

inputs. Supplemental Data 2: Model with a stress input on PI3K. Supplemental Data 3: Model with a stress input on PI3K and Akt-pS473. Supplemental Data 4: Model with a stress input on PI3K and Akt-pS473, but Akt-pS473 alone cannot activate mTORC1. Supplemental Data 5: Model with a stress input on PI3K, Akt-pS473, and mTORC1. The latter model is also deposited in the BioModels repository (Chelliah et al, 2015) and assigned the accession number MODEL1902140002.

All data on which the conclusions of this study are based are available from the corresponding authors upon request.

# Supplementary Information

# Acknowledgements

We thank Daryl P Shanley (University of Newcastle upon Tyne), Marti Cadena Sandoval, and Janina Leyk (University Medical Centre Groningen) for critical reading of the manuscript. We thank the members of the Thedieck lab for helpful discussions. We thank Christy Hong for experimental support. K Thedieck was supported by a Rosalind-Franklin-Fellowship of the University of Groningen. CA Opitz, K Thedieck, P Razquin Navas, and S Schäuble acknowledge support from the BMBF e:Med initiative GlioPATH (01ZX1402). C Sers and K Thedieck acknowledge support from the BMBF e:Med initiative MAPTor-NET (031A426A/B). K Thedieck acknowledges support from Stichting TSC Fonds (calls 2015 and 2017) and the German Research Foundation (TH 1358/3-1). A Sadik, CA Opitz, C Sers, I Heiland, and K Thedieck acknowledge support from the MESI-STRAT project, which has received funding from the European Union's Horizon 2020 research and innovation programme under grant agreement No. 754688. E Faessler is supported by the German Bundesministerium für Bildung und Forschung under grant no. 01ZZ1803G as well as by the Deutsche Forschungsgemeinschaft as part of the CRC 1076 AquaDiva. This work is based on computational models which form part of a pending patent application (Publication number WO2012163440).

## Author Contributions

AM Heberle: resources, data curation, formal analysis, validation, investigation, visualization, methodology, and writing—original draft and project administration.
P Razquin Navas: resources, data curation, formal analysis, validation, investigation, visualization, methodology, and writing—original draft and project administration.
M Makkinje-langelaar: data curation, investigation, and methodology.
K Kasack: investigation.
A Sadik: software, investigation, and visualization.
E Faessler: software.
U Hahn: software.
P Marx-Stoelting: writing—review and editing.
CA Opitz: supervision, funding acquisition, investigation, and writing—review and editing.
C Sers: supervision, funding acquisition, investigation, and writing—review and editing.
I Heiland: conceptualization, software, supervision, investigation, and writing—review and editing.
S Schäuble: conceptualization, software, formal analysis, supervision, investigation, methodology, project administration, and writing—review and editing.
K Thedieck: conceptualization, resources, software, formal analysis, supervision, funding acquisition, investigation, visualization, methodology, project administration, and writing—original draft, review, and editing.

## Conflict of Interest Statement

The authors declare that they have no conflict of interest.

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
