## [Reviewer comments · Life Science Alliance]

Life Science Alliance

The PI3K and MAPK/p38 pathways control stress granule assembly in a hierarchical manner.

Alexander Heberle, Patricia Razquin Navas, Miriam makkinje-Langelaar, Katharina Kasack, Ahmed Sadik, Erik Faessler, Udo Hahn, Philip Marx-Stoelting, Christiane Opitz, Christine Sers, Ines Heiland, Sascha Schäuble, and Kathrin Thedieck

DOI: <https://doi.org/10.26508/lsa.201800257>

Corresponding author(s): Kathrin Thedieck, University of Innsbruck; Ines Heiland, University of Tromsø; and Sascha Schäuble,

Review Timeline:

Submission Date:	2018-11-27
Editorial Decision:	2018-11-28
Revision Received:	2019-02-18
Editorial Decision:	2019-02-21
Revision Received:	2019-03-06
Accepted:	2019-03-07

Scientific Editor: Andrea Leibfried

Transaction Report:

Referee #1 Review

Report for Author:

The paper identifies PI3K and p38 MAPK as upstream regulators of mTORC1 activity in response to arsenite stress and demonstrates their requirement for stress granule assembly. The authors propose a hierarchical relationship between PI3K and p38, with PI3K dominant when it is active, but the importance of p38 increasing as PI3K activity declines.

It has previously been established that mTORC1 is required for stress granule assembly in response to arsenite (Fournier et al, 2013; Sfakianos et al, 2018). The current study extends this by identifying PI3K and p38 signaling pathways as important upstream mediators of this action of mTORC1. This is interesting, particularly the finding that p38 can promote stress granule assembly when PI3K activity is low. However, it is unclear how this work relates to previous findings on p38-mediated mTORC1 activation and if it represents a novel mechanism. Therefore, the study seems rather preliminary at this stage.

Major points:

1. The modeling approach that led the authors to focus on p38 was well described and further work in this direction may throw up interesting candidates/pathways in the future. However, with respect to p38, the modeling seems confirmatory as the p38 pathway has previously been shown to be a positive regulator of mTORC1 in response to arsenite, independently of TSC2 (Wu et al., J Biol Chem 286:31501). Furthermore, other studies have also shown that p38 promotes mTORC1 activity in response to various stresses (Li et al. J Biol Chem 278:13663; Cully et al. Mol Cell Biol 30:481). None of these studies are referenced or discussed.

2. Related to point 1, Wu et al have shown that a specific isoform of p38, p38beta, binds to and phosphorylates Raptor to enhance mTORC1 activity in response to arsenite. The authors should check whether this route of p38 signaling to mTORC1 matches their observations. Another mechanism of p38 mediated activation of mTORC1 occurs via MK2 phosphorylation and inhibition of TSC2 in response to anisomycin stress (Li et al. J Biol Chem 278:13663). The data in the current manuscript appears to exclude this possibility as the inhibition of MK2 does not impact on S6K T389 phosphorylation. Also, in all the previous studies, effects of p38 inhibition on S6K T389 phosphorylation are observed even when PI3K is not inhibited. It would seem essential that the authors of the current study clearly establish whether they have uncovered a novel route of p38 signalling to mTORC1 that is distinct from those reported previously. If so, they should provide some characterization of the mechanism involved.

3. The evidence that p38 plays a key role in SG assembly is not particularly strong. Fig. 5G indicates that there is still a significant amount of SGs present in cells after p38 and PI3K inhibition. This suggests that another stress-activated pathway plays a more significant role. This is supported by the observation that the inhibition of p38 by LY2228820 results in only a 40% decrease in the residual S6K T389 phosphorylation that remains after PI3K inhibition (Fig. 5C).

Minor points:

- (i) The corresponding author has previously reported that mTORC1 induction by arsenite stress was still observed in HeLa cells after PI3K or AKT inhibition, leading to the conclusion that redox activation of mTORC1 is independent of PI3K and AKT, but rather mediated by TSC2 downregulation (Cell 154:859). Is there an explanation for the discrepancy? Could it be the cell type - e.g. the balance between PI3K and p38 pathways varies between cell types.
- (ii) The authors use IRS1 phosphorylation as a readout of mTORC1/S6K activity but this is not affected by either PI3K or AKT inhibition (Fig. EV3), even though S6K activity is reduced. What is the explanation for this? Are these sites on IRS1 targeted by a different protein kinase in response to arsenite?
- (iii) Rephrase line 293 - not sure it can be concluded that 'stress-induction of AKT-S473 does not mediate mTORC1 activation'. This conclusion only seems valid under conditions of PI3K inhibition. Similar issue with the subheading on line 281 - mTORC1 activation by stress is only independent of AKT if PI3K is inhibited.
- (iv) Fig. EV2D - only partial graph line showing for carrier.
- (v) Fig. 5E looks like it includes 5 repeats rather than the 3 stated in the legend.

Referee #2 Review

Report for Author:

Heberle et. al. present data suggesting that PI3K and p38 act upstream of mTORC1 to promote stress granule assembly in cells subjected to arsenite-induced oxidative stress. This is based upon the observation that pharmacologic inhibitors of these kinases modestly inhibit arsenite-induced stress granule assembly. In contrast, previous studies have implicated mTORC1 in the inhibition of stress granule assembly in cells subjected to H₂O₂ or selenite (but not arsenite). The finding that mTORC1 has opposite effects on different types of oxidative stressors suggests the involvement of other, unidentified factors. Indeed, arsenite has profound effects on a large number of kinases and phosphatases, making it hard to sort out what is going on using chemical inhibitors whose selectivity is not absolute. It is also not clear whether studies using arsenite are relevant to effects on cancer cells and chemotherapeutic agents as the authors infer.

The authors also present a modeling analysis that appears to be biased towards PI3K-Akt-mTORC1 signaling and does not take into account the many other signaling pathways that modulate stress granule assembly. This is certainly understandable given the complexity of the system, but it is difficult for me to sort out the significance of these results to cancer cells and cancer chemotherapy which is the author's goal.

Minor concern: Fig. 2E needs editing. The blot is not properly cropped to fit the boxes. Please format the blot images to align the boxes.

Referee #3 Review

Report for Author:

In this manuscript, Heberle et al. show that pharmacological inhibitors of several kinase pathways reduce the localization of G3BP1 in arsenite-induced stress granules (SGs). This is shown for inhibitors of TORC1, PI3K, PDK1 and p38 MAPK. The authors investigate in quite some detail signaling pathways that are activated by arsenite, and the interdependence of these signaling pathways.

My main concerns are that the manuscript draws general conclusions from limited observations, that the analysis remains very descriptive and does not identify targets of these signaling pathways that mediate the effect on G3BP1 localization, and that the functional relevance of these findings is not addressed.

specific comments:

The title claims that PI3P and p38 control SG assembly in "cancer cells", yet the authors have shown this really only in one cell line, MCF7. Would other cancer cells also respond in this way? Would non-transformed cells react differently?

Fig. 1, 2, 5: The authors use G3BP1 as a single marker protein for SGs, hence the signaling pathways may affect G3BP1 localization only. They would need to test several SG markers including poly-A RNA in order to draw more general conclusions.

In all SG quantifications, the term "amount of SGs per cell" is used (y-axis). It is unclear what is meant by the "amount" of SGs. Numbers? Signal intensity?

While the western blots are quantified carefully, some controls are missing: level of G3BP1 upon TORC1 inhibition (Fig. 1), level of total Ras (Fig. 2).

The description of the modeling approach (Fig. 3) in the results section is very hard to follow and barely comprehensible to an audience not familiar with this type of approach.

Fig. 4: The part describing Fig. 4A-F is very complicated and does not appear to be relevant. The only informative part is Fig. I-K, yet the effect of the MK2 inhibitor PF3644022 on Akt activations seems to be very small as judged from the western blots.

The authors repeatedly claim that SGs have a survival function, and link their signaling pathway observations to cell survival. However, there is no attempt to actually show that the pathways controlling G3BP1 localization in SGs do affect survival, nor that G3BP1 or SGs more generally are indeed important for survival in this experimental setup. In the literature it is far from clear whether cell survival is really a major function of SGs, there are numerous examples where this does not appear to be the case. The authors would need to investigate whether localization of G3BP1 or presence of SGs affects cell survival in their experimental setting, or other functions that SGs may have. Without such data, it remains unclear whether the authors's findings are relevant for cellular physiology.

The authors do not address by which mechanism or targets the signaling pathways affect G3BP1 localization in SGs. Is G3BP1 directly phosphorylated? Are other components of SGs phosphorylated? At this point, the manuscript remains very descriptive.

November 28, 2018

Re: Life Science Alliance manuscript #LSA-2018-00257-T

Prof. Kathrin Thedieck
Universities of Groningen, NL and Oldenburg, D
Department of Pediatrics
Antonius Deusinglaan 1
Groningen 9713 AV
Netherlands

Dear Dr. Thedieck,

Thank you for transferring your manuscript entitled "The PI3K and MAPK/p38 pathways control stress granule assembly in cancer cells." to Life Science Alliance. The manuscript was assessed by expert reviewers at another journal before, and the editors transferred those reports to us with your permission.

The reviewers who evaluated your work before were not convinced that you have found a novel route of p38 signalling to mTORC1 under stress that is distinct from those previously reported, and they noted that the mechanism underlying the reported observations remained unclear. The latter point is not a concern for publication in Life Science Alliance, and we would like to invite you to perform a minor revision for publication of your dataset here. As already discussed with you prior to submission to our journal, we would expect a point-by-point response to all concerns raised and accordingly text changes. We would further appreciate adding an analyses of additional cell lines with differential levels of PI3K activity to address the reviewers' main concern and to better support your main conclusion.

You will be guided to complete the submission of your revised manuscript and to fill in all necessary information. Please note that articles published in LSA can have main and supplementary figures.

The typical timeframe for revisions is three months.

Thank you for this interesting contribution to Life Science Alliance. We are looking forward to receiving your revised manuscript.

Sincerely,

- A letter addressing the reviewers' comments point by point.
- An editable version of the final text (.DOC or .DOCX) is needed for copyediting (no PDFs).
- High-resolution figure, supplementary figure and video files uploaded as individual files: See our detailed guidelines for preparing your production-ready images, <http://life-science-alliance.org/authorguide>
- Summary blurb (enter in submission system): A short text summarizing in a single sentence the study (max. 200 characters including spaces). This text is used in conjunction with the titles of papers, hence should be informative and complementary to the title and running title. It should describe the context and significance of the findings for a general readership; it should be written in the present tense and refer to the work in the third person. Author names should not be mentioned.

B. MANUSCRIPT ORGANIZATION AND FORMATTING:

Full guidelines are available on our Instructions for Authors page, <http://life-science-alliance.org/authorguide>

Hi Kathrin,

I hope all is well!

I just wanted to let you know that I have now received input on the modeling part for your paper. I logged this in our system as official reviewer input, this is why you just received a 'sent out for review' notification via email.

I am copying the input received below. Based on it, I would like to ask you to rewrite the section 'parameter estimation and identifiability analysis' slightly and to deposit the model for re-use by others. Please also address the minor comments and consider adding feedback loops to the model. If the latter proves too difficult, please acknowledge in the ms text that feedback loops could have an additional effect not taken into account in the modeling.

Best wishes,
Andrea

Input received:

I read the paper, focusing in particular on the modeling part. I think that the authors in general well described the assumptions made for building the model, although the model description and assumptions are a bit spread between main text, methods and figures in Expanded View so it took quite some time to find all the pieces.

I also appreciated they performed identifiability analysis (which is very important and often missing), but I think that the section on 'parameter estimation and identifiability analysis' could be written in a more clear way and, since they have distributions for the model parameters, they could even show how the uncertainty on the model parameters would affect the simulation.

Regarding reproducibility it would definitely not be possible to reproduce the results without the model made available to the community (e.g. deposited in BioModels). Model details cannot be derived purely from the text, as they just say that the model is based on ordinary differential equations derived from mass-action kinetics but do not provide details on how it is done and they don't show the differential equations.

My main concern on their strategy to use the model for hypothesis testing is that they only considered the effect of adding the stress as activator of different nodes, but they didn't consider the possibility of including other interactions such as feedback loops. For example, there is a known possible positive feedback loop from Akt to IRS/PI3K as well as a negative feedback loop mediated by mTORC1. By looking at the experimental data in Fig EV4 (or EV6 or EV8, the experimental data are the same), in panel A it shows how the IRS1 activation is delayed with respect to the Akt activation (rising after ~20 min instead of immediately after stress) which could support the presence of a positive feedback loop. If this is the case, the small activation that is visible on the data for Akt (especially on Akt-pS473) even upon AKT inhibitions (with MK-2206 panel C) could be sufficient for a strong activation of IRS1 as effect of the feedback mechanisms. I am not saying that these feedback loops are for sure playing an effect in this context but just that they might be important to test using the model as they could play an important role.

Other minor comments on the figures related to the model and the simulations are:

- Fig EV2B could benefit from improvements in the legend (e.g. color code, meaning of arrows, meaning of different blocks' shapes)

- For figures EV4, EV6 and EV8 it would be easier to visualise in just one panel with stress alone, stress + MK2206 and stress + wortmannin in 3 different colours since it seems that this 3 conditions were modeled together and model simulations are the same in the 3 panels. However the measurements for stress alone (blue dots) seems to be different each time (while the continuous model simulation line is always the same) so it is unclear how they used these data in the model optimisation.

-

Andrea Leibfried, PhD
Executive Editor
Life Science Alliance

The PI3K and MAPK/p38 pathways control stress granule assembly in a hierarchical manner.

Heberle and Razquin Navas et al.

Point-by-Point response to the reviewers:

Referee #1:

The paper identifies PI3K and p38 MAPK as upstream regulators of mTORC1 activity in response to arsenite stress and demonstrates their requirement for stress granule assembly. The authors propose a hierarchical relationship between PI3K and p38, with PI3K dominant when it is active, but the importance of p38 increasing as PI3K activity declines.

It has previously been established that mTORC1 is required for stress granule assembly in response to arsenite (Fournier et al, 2013; Sfakianos et al, 2018). The current study extends this by identifying PI3K and p38 signaling pathways as important upstream mediators of this action of mTORC1. This is interesting, particularly the finding that p38 can promote stress granule assembly when PI3K activity is low. However, it is unclear how this work relates to previous findings on p38-mediated mTORC1 activation and if it represents a novel mechanism. Therefore, the study seems rather preliminary at this stage.

Response: We thank the reviewer for careful assessment of our work.

We agree that our study extends current knowledge, as we report that PI3K and p38 act in a hierarchical manner to activate mTORC1 under stress, and promote stress granule assembly. Only the discovery of the PI3K-p38 hierarchy enabled us to identify the effect of p38 on mTORC1 and stress granules.

Indeed, three reports in untransformed cells (further discussed below) have linked p38 to mTORC1 activation upon stress, but have found only relatively little follow up. This may be due to the fact that most tissue culture experiments are conducted in cancer cell lines with hyperactive PI3K, in which the impact of p38 is not observable. Our study reconciles this seeming discrepancy and highlights p38's importance for mTOR signaling, which has been probably so far underestimated.

We address all concerns of the reviewer in detail below.

Major points:

1. The modeling approach that led the authors to focus on p38 was well described and further work in this direction may throw up interesting candidates/pathways in the future. However, with respect to p38, the modeling seems confirmatory as the p38 pathway has previously been shown to be a positive regulator of mTORC1 in response to arsenite, independently of TSC2 (Wu et al., J Biol Chem 286:31501). Furthermore, other studies have also shown that p38 promotes mTORC1 activity in response to various stresses (Li et al. J

Biol Chem 278:13663; Cully et al. Mol Cell Biol 30:481). None of these studies are referenced or discussed.

Response: We are glad to hear that our modeling approach is well described and that referee #1 followed our argumentation. While some of the mechanisms underlying the single steps in the PI3K-p38-mTORC1 signaling network have indeed been reported earlier, we are the first to integrate them into a comprehensive network by dynamic modeling. Only this approach allows to identify the consequences of the dynamic interplay of the molecular mechanisms, i.e. the hierarchy in PI3K and p38 effects on mTORC1 and stress granules.

We thank the reviewer for pointing out the three studies on p38 links to mTORC1, and we introduce this information as follows: *“How does p38 activate mTORC1? So far two mechanisms have been proposed: (i) p38 has been suggested to activate mTORC1 through MK2 (Cully et al., 2010; Li et al., 2003). Yet, in our hands MK2 inhibition did not affect mTORC1 activity (Figs 4I-K and S13I-K), indicating that mTORC1 activation by p38 was MK2-independent; (ii) p38 has been proposed to directly phosphorylate raptor and activate mTORC1 (Wu et al., 2011), but it is unknown whether this occurs in a PI3K dependent manner. Hence, in cells with low PI3K activity, p38 might activate mTORC1 either through raptor or via a hitherto unknown mechanism, which requires further investigation in the future.”*

2. Related to point 1, Wu et al have shown that a specific isoform of p38, p38beta, binds to and phosphorylates Raptor to enhance mTORC1 activity in response to arsenite. The authors should check whether this route of p38 signaling to mTORC1 matches their observations. Another mechanism of p38 mediated activation of mTORC1 occurs via MK2 phosphorylation and inhibition of TSC2 in response to anisomycin stress (Li et al. J Biol Chem 278:13663). The data in the current manuscript appears to exclude this possibility as the inhibition of MK2 does not impact on S6K T389 phosphorylation. Also, in all the previous studies, effects of p38 inhibition on S6K T389 phosphorylation are observed even when PI3K is not inhibited. It would seem essential that the authors of the current study clearly establish whether they have uncovered a novel route of p38 signalling to mTORC1 that is distinct from those reported previously. If so, they should provide some characterization of the mechanism involved.

Response: As pointed out in our response to comment 1, we agree with the reviewer that MK2 does not mediate p38 signals to mTORC1 in our system. As for raptor phosphorylation by p38, Wu et al. (Wu et al., 2011, J Biol Chem 286:31501; PMID: 21757713) indeed reported that direct phosphorylation of Raptor-S863 by p38 contributes to mTORC1 activation by arsenite stress. Yet, this notion has been challenged more recently by a study of Yuan et al. (Genes and Dev., 2015; PMID: 26588989) which used Raptor knock-in mutants to demonstrate that Raptor-S863 phosphorylation inactivates mTORC1. Hence, we cannot exclude that p38 signals through raptor to activate mTORC1, but it might also act through another hitherto unknown mechanism. While we agree that this will be an important matter to be addressed by future research, we consider it beyond the scope of the present study whose major novelty is the discovery of the hierarchy between PI3K and p38 in enhancing mTORC1 and stress granule assembly.

To strengthen this point and show that the hierarchy between PI3K and p38 is a general feature of stress signaling to mTORC1, we have generated new data in further cell lines (Figs 5G and S14D-G), out of which three are of tumour origin (HeLa, CAL51, LN18), and one is a non-malignant cell line (HEK293T). In cells with enhanced PI3K activity upon stress

(CAL51, LN18, HEK293T), PI3K inhibition was required to observe a p38 effect on mTORC1. In HeLa cells PI3K was not responsive to stress, and here p38 inhibition reduced mTORC1 activity also without additional PI3K inhibition. Thus, p38 inhibition directly affects mTORC1 in cells with low PI3K activity; while in cells with high PI3K, the impact of p38 becomes only observable when PI3K is suppressed.

This is in line with the reviewer's comment that in some cells p38's effect on mTORC1 is observable without prior inhibition of PI3K, namely when its activity is low already. Of note, one of the papers the reviewer mentions observed p38 effects in HEK293T cells. In contrast to this study, we observed p38 effects on mTORC1 in HEK293T cells only upon prior inhibition of PI3K.

3. The evidence that p38 plays a key role in SG assembly is not particularly strong. Fig. 5G indicates that there is still a significant amount of SGs present in cells after p38 and PI3K inhibition. This suggests that another stress-activated pathway plays a more significant role. This is supported by the observation that the inhibition of p38 by LY2228820 results in only a 40% decrease in the residual S6K T389 phosphorylation that remains after PI3K inhibition (Fig. 5C).

Note: during the revisions, the figure number changed from **Fig. 5G** to **Fig 5I**.

Response: p38 inhibition significantly reduces the amount of stress granules by over 30% (**Fig. 5I**), and thus we maintain that p38 has a significant role in the promotion of stress granule formation when PI3K activity is low.

Of course this does not exclude the contribution of further mechanisms, and we agree with the reviewer that there are remaining stress granules (**Fig. 5I**) and mTORC1 activity (**Fig. 5C**) when PI3K and p38 are inhibited. This effect could be for instance mediated by inhibition of the mTORC1 repressor TSC2, whose levels are reduced by arsenite exposure (**Fig. S1C and D**), independently of the PI3K-Akt signaling axis (**Fig. S3**).

Minor points:

(i) The corresponding author has previously reported that mTORC1 induction by arsenite stress was still observed in HeLa cells after PI3K or AKT inhibition, leading to the conclusion that redox activation of mTORC1 is independent of PI3K and AKT, but rather mediated by TSC2 downregulation (Cell 154:859). Is there an explanation for the discrepancy? Could it be the cell type - e.g. the balance between PI3K and p38 pathways varies between cell types.

Response: In keeping with our observations in the present study in several cell types, we indeed also observed in an earlier study on mTORC1 inhibition by stress granules that in HeLa cells mTORC1 is stress-inducible when PI3K is inhibited (Thedieck et al., Cell, 2013, Fig. S7F; PMID: 23953116). As pointed out above and correctly observed by our reviewer, we found TSC2 levels reduced upon arsenite exposure in the present (**Fig S1C and D**) and our previous works (Thedieck et al., Cell, 2013, Fig. S7G), which lead us to suggest TSC2 reduction as a possible mechanism of mTORC1 activation under stress.

In the present study, we report that p38 is a major mediator of stress-induction of mTORC1 when PI3K activity is low. This finding complements our earlier study and is not a

discrepancy, as it points to p38 as an important mechanism, which mediates stress inducibility of mTORC1 when PI3K is low. While p38 is important, we do not exclude other mechanisms that might add to mTORC1 inducibility by stress, such as TSC2.

We agree with the reviewer that differences in PI3K activity between different cell types account for differences in the contribution of the PI3K-Akt axis to mTORC1 activation by stress. In fact, HeLa cells, which we used in Thedieck et al. (Cell, 2013) are not PIK3CA transformed (see e.g. Arjumand et al., 2016, Oncotarget; PMID: 27489350). As we show in our new data, HeLa cells exhibit very low PI3K inducibility in comparison to other cell types including MCF-7 (**Figs 5G and S14D-G**). In agreement, p38 inhibition in HeLa reduced mTORC1 activity also without additional PI3K inhibition. Thus, the low PI3K-activity in HeLa renders PI3K-independent stress-signaling branches particularly prominent in this cell type. Our present study complements our earlier findings and explains why stress signaling to mTORC1 is perceived as PI3K-dependent or independent in different cell types.

(ii) The authors use IRS1 phosphorylation as a readout of mTORC1/S6K activity but this is not affected by either PI3K or AKT inhibition (Fig. EV3), even though S6K activity is reduced. What is the explanation for this? Are these sites on IRS1 targeted by a different protein kinase in response to arsenite?

Note: during the revisions, the figure number changed from **Fig. EV3** to **Fig S3**.

Response: Indeed, we show IRS1 phosphorylation in the datasets that were used for initial model parameterization, and the reviewer is correct that IRS1 phosphorylation is not affected by PI3K or Akt inhibition, although S6K phosphorylation is reduced (**Fig S3**). This is in agreement with our subsequent observation (**Fig 2E and F**) that IRS1 does not contribute to mTORC1 activation by arsenite stress, which suggests that mTORC1 is also uncoupled from negative, S6K-IRS1-mediated feedback signaling.

As we demonstrate that IRS1 does not play a role for mTORC1's response to arsenite stress, we have not added further data on its regulation to our manuscript. As we show throughout, p70-S6K-pT389 is reduced but not fully abolished by PI3K and Akt inhibition. Thus, the remaining p70-S6K activity could in principle be sufficient to sustain IRS1 phosphorylation. Interestingly, ERK has been proposed as another kinase, which can phosphorylate IRS1 at the sites monitored here (Luo et al., 2007, Endocrinology; PMID: 17640984). We agree with the reviewer that this observation is interesting. However, as stress-activation of mTORC1 is independent of both IRS1 and the negative feedback loop (**Fig 2E and F**), we consider that analyzing the kinase, which phosphorylates IRS1 upon stress is beyond the scope of our manuscript.

(iii) Reword line 293 - not sure it can be concluded that 'stress-induction of AKT-S473 does not mediate mTORC1 activation'. This conclusion only seems valid under conditions of PI3K inhibition. Similar issue with the subheading on line 281 - mTORC1 activation by stress is only independent of AKT if PI3K is inhibited.

Response: We agree with Referee #1, and we rephrased the statements to "*stress-induction of Akt-pS473 does not mediate mTORC1 activation, when PI3K is inactive*" and "*mTORC1 activation by stress is independent of AKT, when PI3K is inactive*", respectively.

(iv) Fig. EV2D - only partial graph line showing for carrier.

Note: during the revisions, the figure number changed from **Fig. EV2D to Fig S2D**.

Response: In this figure, we show the slope in the area in which TSC2-pT1462 (substrate of Akt) exhibits a linear increase in response to arsenite exposure. By comparing it to the slope of TSC2-pT1462 upon MK-2206 (Akt inhibitor), we quantified the extent of Akt inhibition. To clarify this better, we have included the whole graph line and highlight in red the slope used for the calculation.

(v) Fig. 5E looks like it includes 5 repeats rather than the 3 stated in the legend.

Note: during the revisions, the figure number changed from **Fig. 5E to Fig 5F**.

Response: We thank Referee #1 to indicate this mistake. We have corrected this in the figure legend. We also carefully checked again all other legends and ensure that they are correct.

Referee #2:

1) Heberle et. al. present data suggesting that PI3K and p38 act upstream of mTORC1 to promote stress granule assembly in cells subjected to arsenite-induced oxidative stress. This is based upon the observation that pharmacologic inhibitors of these kinases modestly inhibit arsenite-induced stress granule assembly.

Response: While the PI3K inhibitor Wortmannin significantly reduces the amount of stress granules by more than half (**Fig. 2I and J**), the p38 inhibitor LY2228820 significantly reduces the amount of the remaining stress granules by more than 30% (**Fig. 5H and I**). Thus, PI3K and p38 drive the formation of the majority of stress granules.

2) In contrast, previous studies have implicated mTORC1 in the inhibition of stress granule assembly in cells subjected to H₂O₂ or selenite (but not arsenite). The finding that mTORC1 has opposite effects on different types of oxidative stressors suggests the involvement of other, unidentified factors.

Response: Indeed, the Anderson lab has reported that so-called non-canonical stress granules, induced by selenite (Fujimura et al., 2012, NAR; PMID: 22718973) or H₂O₂ (Emara et al., 2012, BBRC; PMID: 22705549), require 4EBP1. While Fujimura et al. report a reduction in 4EBP1 and S6 phosphorylation by selenite and thereby draw a correlation to mTORC1 inhibition, Emara et al. do not show mTORC1 readouts upon H₂O₂ exposure. However, in our previous work we used the same H₂O₂ concentration as Emara et al. (2 mM), and observed strong activation of mTORC1 (Thedieck et al., Cell 2013, Fig. S7D; PMID: 23953116). Hence, 4EBP1-mediated stress granule induction seems to happen under net inhibition or net activation of mTORC1, and thus might be mediated by 4EBP1 upstream inputs other than mTORC1 (see e.g., Herbert et al., 2002, JBC, PMID: 11799119; Qin et al., 2016, Cell Cycle, PMID: 26901143).

In contrast, we (this manuscript) as well as Sfakianos et al. (2018, CD&D; PMID: 29523872) and Fournier et al. (2013, MCB; PMID: 23547259) showed with mTORC1 inhibitors and/or knockdowns that stress granule assembly upon stressors as diverse as arsenite, FL3 (eIF4A helicase inhibitor), and bortezomib (proteasome inhibitor) require mTORC1 activity. Thus, we propose that mTORC1 has a major role in promoting stress granule assembly. Therefore, the identification of mTORC1's upstream activators under stress, as done here, is key to mechanistically unravel the signaling network mediating stress granule formation.

3) Indeed, arsenite has profound effects on a large number of kinases and phosphatases, making it hard to sort out what is going on using chemical inhibitors whose selectivity is not absolute.

Response: We agree that arsenite has multiple effects. Given the profound importance of mTORC1 for stress granule assembly in response to this and other stressors (see response to comments 1 and 2 of this reviewer) and the wide use of arsenite in the stress granule field, arsenite is a suitable tool compound to unravel the network upstream of mTORC1, which mediates stress granule assembly.

We further agree that chemical inhibitors are often not absolutely selective. To address this issue, we always either used several different inhibitors of the same target, or RNA interference in addition to the inhibitors.

4) It is also not clear whether studies using arsenite are relevant to effects on cancer cells and chemotherapeutic agents as the authors infer.

Response: As pointed out above (response to comment 1-3), mTORC1 is a major driver of stress granule assembly in response to a variety of stresses including arsenite. Thus, arsenite is a suitable tool compound to study stress granule drivers upstream of mTORC1.

The relevance for cancer comes from the fact that many mediators of mTORC1 activity are targeted or are in clinical trials for cancer therapy. We demonstrate here that they have effects on stress granules, which have been hitherto not appreciated, and we propose that these effects may contribute to drug action.

Furthermore, we show in a large breast cancer cohort (TCGA wide) that, as predicted by our computational model and *in vitro* data, correlation of mTORC1 activity with p38 occurs only in tumors with low PI3K activity (**Fig. 6B**). We believe that this is a strong argument in favour of the *in vivo* (human patient) relevance of our data.

5) The authors also present a modeling analysis that appears to be biased towards PI3K-Akt-mTORC1 signaling and does not take into account the many other signaling pathways that modulate stress granule assembly.

Response: As pointed out above (responses to comments 1-4), mTORC1 is a pro-stress-granule-kinase, and we use computational modeling for hypothesis building in order to guide experiments to unravel the signaling cues upstream of mTORC1 which drive stress granule assembly.

As discussed in response to minor point (i) of reviewer #1, this of course does not exclude the involvement of further signaling cues in stress granule formation.

6) This is certainly understandable given the complexity of the system, but it is difficult for me to sort out the significance of these results to cancer cells and cancer chemotherapy which is the author's goal.

Response: This reiterates the argument made in comment 4, and we refer to our response there.

We would also like to re-emphasize that the main focus of the present study is to identify signaling cues upstream of mTORC1 that drive stress granule assembly. In addition, we also point to the medical relevance of our findings, yet, the clinical dimension is not the prime goal of our study but will require further testing in preclinical and clinical settings in future studies. We have detailed this in the last paragraph of our discussion.

Referee #3:

In this manuscript, Heberle et al. show that pharmacological inhibitors of several kinase pathways reduce the localization of G3BP1 in arsenite-induced stress granules (SGs). This is shown for inhibitors of TORC1, PI3K, PDK1 and p38 MAPK. The authors investigate in quite some detail signaling pathways that are activated by arsenite, and the interdependence of these signaling pathways.

Response: We thank referee #3 for his/her positive assessment.

My main concerns are that the manuscript draws general conclusions from limited observations,

Response: As detailed below (answer to comment 1), we added data on further cancer and non-cancer cell lines, as requested by the reviewer.

that the analysis remains very descriptive and does not identify targets of these signaling pathways that mediate the effect on G3BP1 localization,

Response: As described below in more detail (response to comment 8), the pathways, which mediate stress granule assembly downstream of mTORC1 have been described elsewhere, and those papers are cited in our manuscript.

and that the functional relevance of these findings is not addressed.

Response: As indicated below (response to comment 7), the general relevance of stress granules for cellular survival has been shown by us and others elsewhere (Thedieck et al., 2013, Cell; PMID: 23953116; Arimoto et al., 2008, NCB; PMID: 18836437) and is generally accepted in the field (reviewed in detail by Kedersha and Anderson, 2013, TBS; PMID: 24029419).

Furthermore, as detailed above in response to reviewer #2 (comment 4), we demonstrate the potential clinical relevance of our findings in a large breast cancer cohort (TCGA wide), in which we find the same correlation between mTORC1 and p38 readouts with PI3K activity, as suggested by our *in silico* and *in vitro* data with arsenite.

specific comments:

1.) The title claims that PI3P and p38 control SG assembly in "cancer cells", yet the authors have shown this really only in one cell line, MCF7. Would other cancer cells also respond in this way? Would non-transformed cells react differently?

Response: We thank our reviewer for pointing this out.

As indicated in our response to reviewer #1 (point 2), we have included data for four additional transformed and non-transformed cell lines with differences in PI3K activity. PI3K and p38 mediate mTORC1 activation both in transformed and non-transformed cells. Hence,

we conclude that the hierarchy between PI3K and p38 mediated stress-activation of mTORC1 is a general mechanism preserved among different cancer-derived and non-malignant cells.

We agree that the previous title was misleading, as it did not cover the full scope of our discovery. We have therefore changed our title and omitted "in cancer cells".

2.) Fig. 1, 2, 5: The authors use G3BP1 as a single marker protein for SGs, hence the signaling pathways may affect G3BP1 localization only. They would need to test several SG markers including poly-A RNA in order to draw more general conclusions.

Response: G3BP1 is a *bona fide* stress granule marker, which is used throughout in the field (see e.g. Kedersha and Anderson, 2007, ME; PMID: 17923231). The punctuate cytoplasmic pattern, which we observe points to a localization of G3BP1 in stress granules and is to the best of our knowledge in agreement with the vast majority of reports on stress granules (see e.g. examples in Sfakianos et al., 2018, CD&D; PMID: 29523872, Emara et al., 2012, BBRC; PMID: 22705549; Thedieck et al., Cell 2013; PMID: 23953116).

3.) In all SG quantifications, the term "amount of SGs per cell" is used (y-axis). It is unclear what is meant by the "amount" of SGs. Numbers? Signal intensity?

Response: We have counted the number of SGs per cell and we added this information to the manuscript. The legend now reads: "*number of SGs per cell normalized to the arsenite condition*".

4.) While the western blots are quantified carefully, some controls are missing: level of G3BP1 upon TORC1 inhibition (Fig. 1), level of total Ras (Fig. 2).

Response: We have so far not observed effects of mTORC1 inhibition on G3BP1 levels (see e.g. Thedieck et al., Cell, 2013, Fig. 6D; PMID: 23953116).

For completeness, we have included the information also in this manuscript: we added new G3BP1 data to **Fig 1D and S1A and B**, which confirm that the levels remain constant upon arsenite treatment and mTOR inhibition. We introduced the following text:

"Arsenite exposure did not affect G3BP1 levels (Fig S1A and B), indicating that the differences in immunofluorescence were due to granule localization."

"We found that both everolimus and AZD8055 decreased the numbers of G3BP1-positive foci without affecting G3BP1 levels, suggesting that stress granule formation was inhibited (Fig 1D, F and G)".

We also included data on Ras levels, which remained constant upon arsenite treatment (**Fig S1E and F**). We added the following text: "*Arsenite stress enhanced RAS-GTP levels, as determined by increased RBD-bound RAS (Fig 2G and H) while total RAS remained constant (Fig S1E and F)*".

5.) The description of the modeling approach (Fig. 3) in the results section is very hard to follow and barely comprehensible to an audience not familiar with this type of approach

Response: While reviewer #1 found our modeling approach well described, an audience not from the field might indeed find it challenging to understand the technicalities.

To render the modelling approach more accessible to a wider audience, we have revised the method section and included additional supplementary information (**Data S7**) to describe the computational methods in more detail.

6.) Fig. 4: The part describing Fig. 4A-F is very complicated and does not appear to be relevant.

Response: In the current literature, Akt phosphorylated at S473 is widely considered as the main mTORC2 effector (see e.g. Dibble and Cantley, 2015, TCB; PMID: 26159692). Furthermore, Akt-pS473 is often suggested to be upstream of mTORC1 (see e.g. Janku, 2018, Nat. Rev.; PMID: 29508857; Takei and Nawa, 2014, FMN; PMID: 24795562), although this has so far not been shown (see e.g. Jacinto et al., 2006, Cell; PMID: 16962653). Thus, our findings that when PI3K activity is low, Akt-S473 is neither phosphorylated by mTORC2 nor upstream of mTORC1 are of great interest to the mTOR field. Therefore, we decided to include these findings in the main figures.

The only informative part is Fig. I-K, yet the effect of the MK2 inhibitor PF3644022 on Akt activations seems to be very small as judged from the western blots.

Response: We are surprised by this comment as we observe under PI3K inhibition that the MK2 inhibitor PF3644022 reduces Akt-pS473 by ca. 75% (**Fig. 4K**). This has been quantified across four biological replicates.

7.) The authors repeatedly claim that SGs have a survival function, and link their signaling pathway observations to cell survival. However, there is no attempt to actually show that the pathways controlling G3BP1 localization in SGs do affect survival, nor that G3BP1 or SGs more generally are indeed important for survival in this experimental setup. In the literature it is far from clear whether cell survival is really a major function of SGs, there are numerous examples where this does not appear to be the case. The authors would need to investigate whether localization of G3BP1 or presence of SGs affects cell survival in their experimental setting, or other functions that SGs may have. Without such data, it remains unclear whether the authors's finding are relevant for cellular physiology.

Response: As pointed out above, the role of stress granules for cellular survival under a number stresses is well established by us (Thedieck et al. 2013, Cell, Fig 7F-H; PMID: 23953116) and others (see e.g. Arimoto et al., 2008, NCB; PMID: 18836437) and is generally accepted in the field (reviewed in detail by Kedersha and Anderson, 2013, TBS; PMID: 24029419). Therefore, we have opted to not address this point again in this manuscript.

Indeed, so-called non-canonical stress granules induced by selenite have been reported to be pro-apoptotic (Fujimura et al., 2012, NAR; PMID: 22718973), which seems to be in

contrast to stress granules induced by other stresses. While this is certainly an interesting aspect of stress granule biology, we consider the topic of non-canonical stress granules beyond the scope of the present study and the field will need to address this matter in future studies.

8.) The authors do not address by which mechanism or targets the signaling pathways affect G3BP1 localization in SGs. Is G3BP1 directly phosphorylated? Are other components of SGs phosphorylated? At this point, the manuscript remains very descriptive.

Response: mTORC1-dependent stress granule formation is directly mediated by the two major mTORC1 substrates 4EBP1 and S6K, and the related papers are cited in our manuscript. Fournier et al. (2013, MCB; PMID: 23547259) showed that mTORC1-mediated phosphorylation of 4EBP1 allows eIF4F formation, which is required for stress granule assembly. Sfakianos et al. (2018, CD&D; PMID: 29523872) showed that S6K downstream of mTORC1 mediates eIF2alpha phosphorylation, enhancing stress granule assembly.

We use G3BP1 as a *bona fide* stress granule marker, generally applied in the field (see e.g. Kedersha and Anderson, 2007, ME; PMID: 17923231), and we do not imply that mTORC1 might control stress granules by directly impinging on G3BP1.

Referee #4:

I read the paper, focusing in particular on the modeling part. I think that the authors in general well described the assumptions made for building the model, although the model description and assumptions are a bit spread between main text, methods and figures in Expanded View so it took quite some time to find all the pieces. I also appreciated they performed identifiability analysis (which is very important and often missing), but I think that the section on 'parameter estimation and identifiability analysis' could be written in a more clear way and, since they have distributions for the model parameters, they could even show how the uncertainty on the model parameters would affect the simulation.

Response: We agree with the reviewer that identifiability analysis is an important aspect of model analysis. As not all parameters were identifiable, we underwent an iterative process combining parameter identifiability, repeated parameter sampling from different starting points in the solution space (latin hypercube sampling and gradient descent) and clustering for parameter values that best support our experimental data. We have rewritten the methods section on 'parameter estimation and identifiability analysis' to clarify this procedure. Furthermore, we added a supplemental document explaining in more detail the iterative parameter estimation method which we used (**Data S7**). The ranges we provide for the parameters in the supplementary material reflect the initial parameter value ranges, but not the parameter distributions that support equally likely model parameters according to the AIC calculations. Therefore, the given parameter ranges do not provide uncertainty of model parameters and the reviewers advise "to show how the uncertainty on the model parameters would affect the simulation" cannot be implemented. We regret this misconception and hope that the added text and supplementary data clarify that different parameter distributions resulting from repeated sampling runs were used.

Regarding reproducibility it would definitely not be possible to reproduce the results without the model made available to the community (e.g. deposited in BioModels). Model details cannot be derived purely from the text, as they just say that the model is based on ordinary differential equations derived from mass-action kinetics but do not provide details on how it is done and they don't show the differential equations.

Response: As for all our modeling work, we make the models available as supplementary data (**Data S1-S5**). In addition, the final model (model version V) is deposited in BioModels (accession number: MODEL1902140002) and will be made publically accessible by the BioModels curators once our manuscript is published.

My main concern on their strategy to use the model for hypothesis testing is that they only considered the effect of adding the stress as activator of different nodes, but they didn't consider the possibility of including other interactions such as feedback loops. For example, there is a known possible positive feedback loop from Akt to IRS/PI3K as well as a negative feedback loop mediated by mTORC1. By looking at the experimental data in Fig EV4 (or EV6 or EV8, the experimental data are the same), in panel A it shows how the IRS1 activation is delayed with respect to the Akt activation (raising after ~20 min instead of immediately after stress) which could support the presence of a positive feedback loop. If this is the case, the small activation that is visible on the data for Akt (especially on Akt-pS473) even upon AKT

inhibitions (with MK-2206 panel C) could be sufficient for a strong activation of IRS1 as effect of the feedback mechanisms. I am not saying that these feedback loops are for sure playing an effect in this context but just that they might be important to test using the model as they could play an important role.

Note: during the revisions, the figure numbers changed from **Fig. EV4, EV6 and EV8 to Fig S4, S6 and S8**, respectively.

Response: There is indeed a negative feedback from mTORC1 to PI3K through IRS1 phosphorylation by S6K (Um et al. 2004, Nature, PMID: 15306821, Carlson et al. 2004, BBRC, PMID: 15020250, Tzatsos et al. 2006, Mol. Cel. Biol., PMID: 16354680). To test if IRS1 plays a role during stress-activation of mTORC1, we inhibited IRS1 by knockdown and monitored mTORC1 activity upon arsenite treatment (**Fig 2E and F**). We found that IRS1 inhibition does not affect mTORC1 signaling dynamics under stress. We thus concluded that mTORC1 stress-activation is independent of IRS1, and this also excludes the involvement of positive or negative feedback loops that act through IRS1.

Nonetheless, for completeness all our models (**Data S1-5**) include this negative feedback loop, which is represented as an inhibitory input from active S6K to IRS1. Thus, our models can in principle account for feedback effects through IRS1. A model without external stress inputs but including the negative feedback loop (**Data S1**) cannot reproduce the dynamics observed experimentally (**Figs S1C and D and S3**). Therefore, in line with our experimental data, our modeling data also suggests that negative feedback through IRS1 cannot explain the dynamics of mTORC1 stress-activation (**Fig 2E and F**).

Other minor comments on the figures related to the model and the simulations are:

- Fig EV2B could benefit from improvements in the legend (e.g. color code, meaning of arrows, meaning of different blocks' shapes)

Note: during the revisions, the figure number changed from **Fig. EV2B to Fig S2B**.

Response: The legend of **Fig S2B** was extended and reads now:

*“Topology of model I without stress input (Data S1). Brown squares = species included in the model, circles = species variants (P, phosphorylation at the indicated site; cyt, cytosolic localization; mem, cell membrane localization; *,active state), dark brown = observable species, species in ellipses = possible inputs to the model (insulin, amino acids) and inhibitory agents (MK-2206, wortmannin), dark blue lines = mTORC2 activity, light blue lines = mTORC1 activity.”*

- For figures EV4, EV6 and EV8 it would be easier to visualise in just one panel with stress alone, stress + MK2206 and stress + wortmannin in 3 different colours since it seems that this 3 conditions were modeled together and model simulations are the same in the 3 panels. However the measurements for stress alone (blue dots) seems to be different each time (while the continuous model simulation line is always the same) so it is unclear how they used these data in the model optimisation.

Note: during the revisions, the figure numbers changed from **Fig. EV4, EV6 and EV8** to **Figs S4, S6 and S8**, respectively.

Response: We agree with the reviewer that Figs S4, S6 and S8 benefit from different colours for different perturbation datasets. Hence, we have changed the figures accordingly and arsenite exposure alone (blue), arsenite stress + MK2206 perturbation (gray), and arsenite stress + wortmannin perturbation (orange) have distinct colours.

We have decided against showing all experiments (stress only, stress + MK2206 and stress + wortmannin) in one panel for two reasons: (i) we want to highlight that each parameterization data set was obtained from separate experiments; and (ii) showing all this information in one panel would increase their complexity and make the content less accessible.

Referee #4 noticed correctly that the measurements for arsenite stress alone (blue dots) are different in each perturbation data set, while the continuous model simulation line is the same for each panel (**Figs S4, S6, S8, S10, S12**). The reason is that the models were calibrated on the timecourse data upon arsenite exposure, shown in **Fig S1C and D**. This condition was also included as a control in the perturbation experiments with MK-2206 (**Fig S3C and D**) or wortmannin (**Fig S3A and B**). Therefore, the control data (blue dots) (**Figs S4, S6, S8, S10, S12**) differ between the datasets in the range of the biological variation, while the simulation for the control condition is the same. We opted to include the simulation for the control condition in all experiments to show the match with the individual control experiments. This demonstrates that the control in the inhibitor experiments would not enforce new parameter calibration.

In order to clarify this better, we have added information to the legends of **Figs S4, S6, S8, S10 and S12**. They read now as follows (example for Fig S4B): *“Dots represent the experimental data in **Fig S3C and D**, shown as mean ± SEM. Lines represent simulated time courses. The simulation of arsenite stress only (blue) is calibrated on the experimental data shown in **Fig S1C and D** and identical to lines shown in **Fig S4A**. The simulation of arsenite + MK-2206 (gray) is calibrated on the experimental data shown in **Fig S3C and D**”.*

February 21, 2019

RE: Life Science Alliance Manuscript #LSA-2018-00257-TR

Prof. Kathrin Thedieck
University of Innsbruck
Institute of Biochemistry
Innrain 80-82
Innsbruck 6020
Austria

Dear Dr. Thedieck,

Thank you for submitting your revised manuscript entitled "The PI3K and MAPK/p38 pathways control stress granule assembly in a hierarchical manner." I appreciate the introduced changes and would be happy to publish your paper in Life Science Alliance pending final revisions necessary to meet our formatting guidelines:

- please ask all corresponding authors to link their profile in our submission system to their ORCID iD, they should have received an email with instructions on how to do so
- please check all author contributions; note that we usually adhere to the ICMJE author contribution guidelines

A. FINAL FILES:

-- Summary blurb (enter in submission system): A short text summarizing in a single sentence the study (max. 200 characters including spaces). This text is used in conjunction with the titles of papers, hence should be informative and complementary to the title. It should describe the context

and significance of the findings for a general readership; it should be written in the present tense and refer to the work in the third person. Author names should not be mentioned.

B. MANUSCRIPT ORGANIZATION AND FORMATTING:

Sincerely,

March 7, 2019

RE: Life Science Alliance Manuscript #LSA-2018-00257-TRR

Prof. Kathrin Thedieck
University of Innsbruck
Institute of Biochemistry
Innrain 80-82
Innsbruck 6020
Austria

Dear Dr. Thedieck,

Thank you for submitting your Research Article entitled "The PI3K and MAPK/p38 pathways control stress granule assembly in a hierarchical manner". It is a pleasure to let you know that your manuscript is now accepted for publication in Life Science Alliance. Congratulations on this interesting work.

DISTRIBUTION OF MATERIALS:

Again, congratulations on a very nice paper. I hope you found the review process to be constructive and are pleased with how the manuscript was handled editorially. We look forward to future exciting submissions from your lab.

Sincerely,

Andrea Leibfried, PhD
Executive Editor
Life Science Alliance
Meyerohofstr. 1
69117 Heidelberg, Germany
t +49 6221 8891 502
e a.leibfried@life-science-alliance.org
www.life-science-alliance.org